# Sample Complexity Bounds for Robust Mean Estimation
# with Mean-Shift Contamination

**Ilias Diakonikolas** [1]  **Giannis Iakovidis** [1]  **Daniel Kane** [2]  **Sihan Liu** [2]

## Abstract

We study the basic task of mean estimation in the presence of mean-shift contamination. In the mean-shift contamination model, an adversary is allowed to replace a small constant fraction of the clean samples by samples drawn from arbitrarily shifted versions of the base distribution. Prior work characterized the sample complexity of this task for the special cases of the Gaussian and Laplace distributions. Specifically, it was shown that consistent estimation is possible in these cases, a property that is provably impossible in Huber's contamination model. An open question posed in earlier work was to determine the sample complexity of mean estimation in the mean-shift contamination model for general base distributions. In this work, we study and essentially resolve this open question. Specifically, we show that, under mild spectral conditions on the characteristic function of the (potentially multivariate) base distribution, there exists a sample-efficient algorithm that estimates the target mean to any desired accuracy. We complement our upper bound with a qualitatively matching sample complexity lower bound. Our techniques make critical use of Fourier analysis, and in particular introduce the notion of a Fourier witness as an essential ingredient of our upper and lower bounds.

## 1. Introduction

Robust statistics (Huber & Ronchetti, 2009) is the field that studies the design of accurate estimators in the presence of data contamination. This study is motivated by a range of practical applications, where the standard i.i.d. assumption holds only approximately. For example, the input data may be systematically manipulated by various sources or include out-of-distribution points; see, e.g., (Barreno et al., 2010; Biggio et al., 2012; Steinhardt et al., 2017; Tran et al., 2018; Diakonikolas et al., 2019a; Du et al., 2024). Originating in the pioneering works of Tukey and Huber (Tukey, 1960; Huber, 1964), the field developed minimax-optimal robust estimators for classical problems. More modern lines of work in computer science have revived robust statistics from an algorithmic viewpoint by providing polynomial-time estimators in high dimensions (Diakonikolas et al., 2019b; Lai et al., 2016; Diakonikolas & Kane, 2023).

The classical contamination model in the field of Robust Statistics, known as Huber's contamination model (Huber, 1964), is defined as follows. For an inlier distribution $D$ over $\mathbb{R}^d$ and a contamination proportion $\alpha \in (0, 1/2)$, each sample $x$ is drawn from $D$ with probability $1 - \alpha$, and from a unknown (potentially arbitrary) distribution $Q$ with probability $\alpha$. Huber's model has been the main focus of prior work in the field, both from the information-theoretic and the algorithmic standpoints. As the model allows for an arbitrary outlier distribution $Q$, it suffers from inherent information-theoretic limitations. Specifically, even for the basic task of estimating the mean of the simplest inlier distributions (e.g., univariate Gaussians), no estimator can achieve error better than $\Omega(\alpha)$—independent of the sample size. That is, Huber's model does not allow for consistent estimators, i.e., estimators whose error becomes arbitrarily small as the sample size increases.

To achieve consistency, some assumptions on the structure of the outlier distribution $Q$ are necessary. In this work, we focus on the basic task of robust mean estimation in a prominent data contamination model, known as the *mean-shift* contamination, in which consistency may be possible. In the mean-shift contamination model, the outlier distribution is assumed to consist of arbitrary mean-shifts of the inlier distribution. Formally, we have the following definition.

**Definition 1.1** (Mean-Shift Contamination Model). Let $\alpha \in (0, 1/2)$, let $D, Q$ be distributions over $\mathbb{R}^d$ with $\mathbf{E}_{x \sim D}[x] = 0$. An $\alpha$-mean-shift contaminated sample from $D_\mu$ is generated as follows:

1. With probability $1 - \alpha$, output a *clean* sample $x = \mu + y$ where $y \sim D$.
2. With probability $\alpha$, an adversary draws a *shift* vector $z$ from $Q$, and outputs an *outlier* sample $x = z + y$ where

Authors are listed in alphabetical order. [1]University of Wisconsin-Madison [2]University of California, San Diego. Correspondence to: Giannis Iakovidis <iakovidis@wisc.edu>.

*Proceedings of the $43^{rd}$ International Conference on Machine Learning*, Seoul, South Korea. PMLR 306, 2026. Copyright 2026 by the author(s).

$$y \sim D.$$

We denote by $D_\mu^{(\alpha)}$ the resulting observed distribution.

The mean-shift contamination model has been intensively studied in recent years, in both the statistics (Cai & Jin, 2010; Collier & Dalalyan, 2019; Carpentier et al., 2021; Kotekal & Gao, 2025) and computer science (Li, 2023; Diakonikolas et al., 2025) literature. Beyond mean estimation, variants of the model have also been considered in the context of regression tasks (Sardy et al., 2001; Gannaz, 2007; McCann & Welsch, 2007; She & Owen, 2011). Historically, this model traces back to the multiple-hypothesis-testing literature, which treats the null parameters as unknown and thus estimates them prior to testing (Efron, 2004; 2007; 2008).

Prior work has studied the sample and computational complexity of mean-shift contamination in two benchmark settings—Gaussian and Laplace base distributions—providing both information-theoretic (Carpentier et al., 2021; Kotekal & Gao, 2025) and algorithmic (Li, 2023; Diakonikolas et al., 2025) results. An immediate corollary of these works is that for these two prototypical base distributions, consistent estimation is indeed possible. Specifically, for the Gaussian case, (Kotekal & Gao, 2025) determined the minimax error rates and (Diakonikolas et al., 2025) gave the first polynomial-time estimator with near-minimax rates in high dimensions.

Despite this recent progress, several fundamental questions regarding estimation in this model remain poorly understood when we depart from the Gaussian/Laplace regimes. *Under what conditions on the distribution $D$ is consistent mean estimation with mean-shift contamination possible? Can we qualitatively characterize the sample complexity of estimating the mean of a general distribution $D$ in this model?* Understanding these questions is of fundamental interest and has been posed as an open problem in prior work. Specifically, the work of (Kotekal & Gao, 2025) states that "an interesting problem is to prove a matching minimax lower bound for estimating the mean with a generic base distribution $D$." As our main contribution, we answer this question by providing a qualitative characterization of the sample complexity of this problem.

### 1.1. Our Results

For a distribution $D$, we define the following quantity:

$$\delta = \delta(\epsilon, \alpha, D) := \inf_{\|v\| \geq \epsilon} \sup_{\omega:\ \mathrm{dist}(\omega \cdot v, \mathbb{Z}) \geq \alpha} |\phi_D(\omega)|. \quad (1)$$

In words, $\delta$ is the worst-case, over all mean-shift directions $v$ with $\|v\| \geq \epsilon$, largest Fourier magnitude $|\phi_D(\omega)|$ over frequencies $\omega$ whose projection $\omega \cdot v$ stays at least $\alpha$ away from every integer. Intuitively, $\{\omega : \mathrm{dist}(\omega \cdot v, \mathbb{Z}) \geq \alpha\}$

*Table 1.* Sample complexity for mean estimation under $\alpha$-mean-shift contamination.

| DISTRIBUTION $D$ | UPPER BOUND | LOWER BOUND ($d=1$) |
|---|---|---|
| $\mathcal{N}(0, I_d)$ | $\widetilde{O}\left(d\, e^{O((\alpha/\epsilon)^2)}\right)$ | $\Omega\left(e^{\Omega((\alpha/\epsilon)^2)}\right)$ |
| $\mathrm{Lap}_d(0, I_d)$ | $\widetilde{O}(d\, \alpha^2/\epsilon^4)$ | $\Omega\left((\alpha/\epsilon)^{1/2}\right)$ |
| $\mathrm{Unif}([-1,1])$ | $\widetilde{O}(1/\epsilon)$ | $\Omega\left((\alpha/\epsilon)^{1/6}\right)$ |
| $\left(\mathrm{Unif}([-1,1])\right)^{*m}$ | $\widetilde{O}\left(\alpha^{-2}\left(O(\alpha/\epsilon)\right)^{2m}\right)$ | $\Omega\left((\alpha/\epsilon)^{(2m-1)/6}\right)$ |

is the set of frequencies which the adversary cannot fully corrupt. As we will establish, the parameter $\delta$ characterizes the sample complexity of estimating the mean $\mu$ of the contaminated distribution $D_\mu^{(\alpha)}$. Particularly, in our sample complexity lower bound (Lemma 4.6), we show that the adversary can zero out the characteristic function $\phi_{D_\mu^{(\alpha)}}$ throughout the set $\mathrm{dist}(\omega \cdot v, \mathbb{Z}) \leq \alpha$. Conversely, in our sample complexity upper bound (Theorem 3.2), we prove that every $\omega$ outside this set is suitable for identifying the difference between $\widehat{\mu}$ and $\mu$ when $\widehat{\mu} - \mu = v$ and $\|v\| = \epsilon$. The difficulty of identifying this difference is governed by the magnitude $|\phi_D(\omega)|$; and since the mean shift could lie in an arbitrary direction $v$, we take the $\inf_v$.

Our main result is informally stated below (see Theorems 3.2 and 4.1 for more detailed formal statements and Remark 4.3 for a comparison of our upper and lower bounds):

**Theorem 1.2** (Informal Main Result). *Let $D$ be a distribution over $\mathbb{R}^d$ with characteristic function $\phi_D$, $\alpha \in (0, 1/2)$ be the contamination parameter, and $\epsilon$ be the target error. Then, under mild technical assumptions on $D$, there exists an algorithm that estimates the mean of $D_\mu$ from $\widetilde{O}_\alpha(d/\delta^2)$ i.i.d. $\alpha$-mean-shift contaminated samples. Moreover, even for $d = 1$, any algorithm requires at least $1/\delta^{\Omega(1)}$ samples.*

A few remarks are in order. First, we note that our univariate sample complexity lower bound can be straightforwardly extended to a $(d/\delta)^{\Omega(1)}$-sample lower bound in $d$ dimensions for product distributions; see Remark 4.2. Second, our Fourier characterization applies to a broad range of base distributions well beyond the Gaussian and Laplace families. We illustrate our results with a few basic examples in Table 1 (see Appendix D).

*Remark* 1.3 (Consistency). Theorem 1.2 yields an easily verifiable condition under which consistency is impossible. Namely, no algorithm can estimate the mean to error $\epsilon$ if the set $\{\omega : \mathrm{dist}(\omega\epsilon, \mathbb{Z}) \geq \alpha\}$ has zero Fourier mass (i.e., if $\delta = 0$ in the theorem's notation). Any distribution with a band-limited characteristic function, i.e., such that there exists $B$ with $\phi(\omega) = 0$ for all $|\omega| \geq B$, satisfies this condition. For example, consider the distribution with density proportional to $\mathrm{sinc}^2(x)$. Note that $\mathrm{sinc}^2$ is the Fourier transform of the triangular window, namely the

function equal to $1 - |\omega|$ for $|\omega| < 1$ and 0 otherwise (i.e., the convolution of two rectangular windows). Since both functions are even, the converse relationship holds as well, which gives us an example of a distribution with band-limited characteristic function.

### 1.2. Notation and Preliminaries

We use $\mathbb{Z}_+$ for the set of positive integers. For $n \in \mathbb{Z}_+$, we denote $[n] := \{1, \ldots, n\}$. For a vector $x$ we denote by $\|x\|$ its Euclidean norm. We define by $\delta_z$ the Dirac measure at $z \in \mathbb{R}^d$, i.e., $\delta_z(A) = \mathbb{1}\{z \in A\}$ for all Borel sets $A \subseteq \mathbb{R}^d$. For distributions $P, Q$, we denote by $P * Q$ their convolution. For a distribution $D$ over $\mathbb{R}^d$ and $\mu \in \mathbb{R}^d$, we use $D_\mu$ for the shifted distribution $D * \delta_\mu$. We also denote by $P_z$ the distribution $\delta_z * P$. For $A \subseteq \mathbb{R}^d$ and $\mathbf{x} \in \mathbb{R}^d$, the distance of $\mathbf{x}$ to $A$ is $\mathrm{dist}(\mathbf{x}, A) = \min_{\mathbf{y} \in A} \|\mathbf{x} - \mathbf{y}\|$. Let $\mathcal{B}_d(R) := \{\theta \in \mathbb{R}^d : \|\theta\| \leq R\}$ be the Euclidean ball of radius $R$ on $\mathbb{R}^d$. For a measurable function $f : \mathbb{R} \to \mathbb{C}$ and $1 \leq p < \infty$, define $\|f\|_{L^p} := \left( \int_{\mathbb{R}} |f(x)|^p, dx \right)^{1/p}$, with value $+\infty$ if the integral diverges. Denote by $(h(x))_+ := \max(h(x), 0)$ and by $(h(x))_- := \max(-h(x), 0)$. Denote by $\mathrm{sinc}$ the function defined as $\mathrm{sinc}(x) := \sin(\pi x)/(\pi x)$ for $x \neq 0$ and $\mathrm{sinc}(0) = 1$.

We use $a \lesssim b$ to denote that there exists an absolute constant $C > 0$ (independent of the variables or parameters on which $a$ and $b$ depend) such that $a \leq Cb$.

**Definition 1.4** (Characteristic function). Let $P$ be a distribution over $\mathbb{R}^d$. We define its characteristic function by $\phi_P(\omega) := \mathbf{E}_{x \sim P} \left[ e^{2\pi i(\omega \cdot x)} \right], \omega \in \mathbb{R}^d$. Note that the characteristic function is the Fourier transform of the underlying probability measure.

## 2. Our Techniques

**Upper Bound** In the mean-shift contamination model, the observed distribution is a convolution of the form $D_\mu^{(\alpha)} = D * Q$, where $D$ is the base distribution (centered at 0) and $Q$ is the distribution over the mean vector (designed by an adversary) that is guaranteed to put at least $(1 - \alpha)$ mass on the target clean mean vector $\mu$. This structure naturally invites Fourier analysis, yielding the characteristic function identity

$$\phi_{D_\mu^{(\alpha)}}(\omega) = \phi_D(\omega) \, \phi_Q(\omega).$$

Conveniently, $\phi_{D_\mu^{(\alpha)}}(\omega)$ can be approximated using samples and $\phi_D(\omega)$ has a known functional form, thereby allowing us to approximate $\phi_Q(\omega)$ via the ratio. This deconvolution step is the reason Fourier analysis is central, the known factor $\phi_D(\omega)$ is common to both the clean and outlier parts. Thus, after dividing by $\phi_D(\omega)$, the problem becomes one of finding frequencies where the phase of an incorrect candidate $\widehat{\mu}$ differs noticeably from that of the true mean $\mu$, while $|\phi_D(\omega)|$ remains large enough for the division to be

statistically stable. Specifically, if $Q$ were a point mass on $\mu$, its characteristic function would be exactly $\exp(2\pi i\mu \cdot \omega)$. By the assumption of the contamination model, $Q$ has to put at least $(1 - \alpha)$ mass on the clean mean vector $\mu$. As a result, it is not hard to show that $\phi_Q(\omega)$ is pointwise $2\alpha$ close to $\exp(2\pi i\mu \cdot \omega)$. Next, we proceed to show how the above observations allow us to distinguish between the cases where a candidate vector $\widehat{\mu}$ is close to the clean vector $\mu$ or is at least $\epsilon$ far from $\mu$. If we can do so, then we can solve the mean estimation task by scanning over all candidate mean vectors lying in a sufficiently fine cover.

Towards distinguishing whether $\widehat{\mu}$ is close to $\mu$ or far from it, we want to decide whether $\phi_Q(\omega)$ is close to $\exp(2\pi i\widehat{\mu} \cdot \omega)$ (as it would be if $Q$ placed most of its mass near $\widehat{\mu}$) or noticeably different. For this, it suffices to find a *witness* frequency $\omega^*$ for which the difference $|\exp(2\pi i\widehat{\mu} \cdot \omega^*) - \exp(2\pi i\mu \cdot \omega^*)|$ is large. This happens whenever $(\widehat{\mu} - \mu) \cdot \omega^*$ is far from an integer. Since $(\widehat{\mu} - \mu) \cdot \omega^* \leq \|\widehat{\mu} - \mu\|\|\omega\|$; for close candidates this requires a very large $\|\omega^*\|$, whereas for $\epsilon$-far candidates, choosing $\|\omega^*\|$ on the order of $1/\epsilon$ suffices.

Once we have such a witness, it remains to construct a sufficiently good estimate of $\phi_Q(\omega^*)$. Our algorithm tries to estimate $\phi_Q$ via the ratio form $\phi_{D_\mu^{(\alpha)}}/\phi_D$. Thus, for the estimator to have reasonable concentration, the denominator $\phi_D$ at $\omega^*$ necessarily needs to be bounded away from 0. In particular, provided that the witness satisfies that $(\widehat{\mu} - \mu) \cdot \omega^*$ is at least $A$-far from integers and $\phi_{D_\mu^{(\alpha)}}(\omega^*)$ is at least $\delta$, we can estimate $\phi_{D_\mu^{(\alpha)}}$—and consequently $\phi_Q(\omega^*)$—up to sufficient accuracy with roughly $O(A^{-1}\delta^{-2})$ samples.

In summary, in order to learn the mean vector $\mu$ up to $\epsilon$ accuracy, all we need is that for any error vector $v = \hat{\mu} - \mu$ of $\ell_2$ norm at least $\epsilon$, there exists a witness frequency $\omega_v$ such that (i) $v \cdot \omega_v$ is bounded away from integers, and (ii) $\phi_{D_\mu^{(\alpha)}}(\omega_v)$ is bounded away from 0. We call this the frequency-witness condition of the distribution $D$ and its formal definition can be found in Definition 3.1.

**Lower Bound** Our lower bound shows that an $L^2$ version of the above frequency witness condition is essentially necessary for mean estimation under the mean-shift contamination model. Consider the basic case of univariate mean estimation. In this case, the negation of the frequency-witness condition with respect to the error $v = \epsilon$ simplifies into saying that the characteristic function restricted on points far from the integer multiples of $\epsilon^{-1}$ have small $L^\infty$ norm, i.e., $\|\phi_D(\omega) \mathbb{1}\{\mathrm{dist}(\epsilon\omega, \mathbb{Z})\}\|_{L^\infty} \leq \delta$. In Theorem 4.1, we show that if the base distribution $D$ satisfies an $L^2$ version of the above bound, i.e., $\|\phi_D(\omega) \mathbb{1}\{\mathrm{dist}(\epsilon\omega, \mathbb{Z}) > \Theta(\alpha)\}\|_{L^2} \leq \delta$, (together with some natural tail bound condition) then $\mathrm{poly}(\delta^{-1})$ many samples are also statistically necessary for mean estimation up to error $\epsilon$ under mean-shift contamination.

At a high level, we would like to design a pair of univariate distributions $Y_1, Y_2$, where $Y_i$ is a univariate distribution that puts at least $(1-\alpha)$ amount of mass on $\pm\epsilon$ respectively, such that the convolutions $Q_i = D * Y_i$ are statistically hard to distinguish, i.e., $\|Q_1 - Q_2\|_{L^1} \leq \text{poly}(\delta)$. Our starting point is the famous Plancherel theorem, which allows us to equate the $L^2$ norm of a function to its Fourier transform. Provided that $Q_1$ and $Q_2$ both satisfy sufficiently strong tail bounds, we can then bound their the total variation distance by their $L^2$ distance, and consequently the $L^2$ distance between their characteristic function $\Delta\phi = \phi_{Q_1} - \phi_{Q_2}$. See Lemma 4.5 for the details of this argument.

To control $\Delta\phi$, we rewrite it as

$$\Delta\phi = \phi_{D*Y_1} - \phi_{D*Y_2}$$
$$= \left((1-\alpha)(e^{\pi i \epsilon \omega} - e^{-\pi i \epsilon \omega}) + \alpha(\phi_{Y_1} - \phi_{Y_2})\right)\phi_D.$$

Note that by our assumption $\phi_D$ already has small $L^2$ norm while restricted on points *far* from integer multiples of $\epsilon^{-1}$. Thus, to ensure that $\Delta\phi$ *globally* has small $L^2$ norm, it suffices to ensure that $\left((1-\alpha)\left(e^{\pi i \epsilon \omega} - e^{-\pi i \epsilon \omega}\right) + \alpha\left(\phi_{Y_1} - \phi_{Y_2}\right)\right)$ has small $L^2$ norm on points *close* to integer multiples of $\epsilon^{-1}$. More precisely, we want to design the pair $Y_1, Y_2$ such that $(\phi_{Y_2} - \phi_{Y_1})$ matches with the Fourier signal $\widehat{f}(\omega) := \alpha^{-1}(1-\alpha)\left(e^{\pi i \epsilon \omega} - e^{-\pi i \epsilon \omega}\right)$ on $\mathbb{1}\{\text{dist}(\omega\epsilon, \mathbb{Z}) < \Theta(\alpha)\}$. It is tempting to consider directly the inverse Fourier transform of $\widehat{f}(\omega)$ and check whether we can write it as a difference of two probability measures. Unfortunately, $\widehat{f}$ can be as large as $2(1-\alpha)/\alpha > 2$, which implies that it cannot be expressed as the difference of any pair of probability measures.

To mitigate the issue, we note that the extreme values of $\widehat{f}$ only occurs periodically at $\omega$ that are far from integer multiples of $\epsilon^{-1}$. Recall that this is precisely the range in which $(\phi_{Y_2} - \phi_{Y_1})$ does not need to approximate $\widehat{f}$. Hence, we can simply truncate $\widehat{f}$ to be 0 at these periodic bands with extreme values. Naively, one would just multiply $\widehat{f}$ with a periodic window function to mask out its extreme values. Yet, doing so would make the resulting function no longer smooth, which is a required condition for its inverse Fourier transform to have rapidly decaying tail bounds. To ensure smoothness, we will further mollify the window function by convolving it with various scaled versions of itself. With some standard Fourier analysis, one can show that the convolved window function has globally bounded derivatives and interpolates smoothly from 0 at points far from the window area to 1 within the window area. Consequently, the derivative bound in the frequency domain gives rise to the desired tail bounds on $Y_2 - Y_1$. The detail of this argument is given in the proof of Lemma 4.6.

## 3. Sample-Efficient Algorithm

In this section, we show that under certain regularity assumptions on the characteristic function of $D$, we can design a sample efficient algorithm. In particular, the assumption that we will use is the following:

**Definition 3.1** (Frequency-witness condition)**.** We say that a distribution $D$ over $\mathbb{R}^d$ satisfies the $(\epsilon, A, \delta)$-frequency-witness condition if: $\forall v \in \mathbb{R}^d$ with $\|v\| \geq \epsilon$ $\exists \omega \in \mathbb{R}^d$ such that $|\sin(\pi v \cdot \omega)| \geq A$, and $|\phi_D(\omega)| \geq \delta$.

Moreover, we call any such $\omega$ an $(\epsilon, A, \delta)$-frequency-witness for the vector $v$.

Denote by $D_\mu^{(\alpha)}$ the $\alpha$-mean-shift contaminated version of $D_\mu$ (see Definition 1.1). Let $\widehat{\mu} \in \mathbb{R}^d$ be a candidate mean. This condition essentially guarantees that every *incorrect* direction $v = \widehat{\mu} - \mu$ with $\|v\| \geq \epsilon$ has an associated frequency $\omega$ where $D$ has nontrivial Fourier mass ($|\phi_D(\omega)| \geq \delta$) and where the phase shift induced by $v$ is *detectable*: $|\sin(\pi \omega \cdot v)| \geq A$. At such a frequency, the population characteristic function of the contaminated model,

$$\phi_{D_\mu^{(\alpha)}}(\omega) = \phi_D(\omega)\left((1-\alpha)e^{2\pi i \omega \cdot \mu} + \alpha \phi_Q(\omega)\right),$$

cannot be well-approximated by $(1-\alpha)\, e^{2\pi i \omega \cdot \widehat{\mu}}\, \phi_D(\omega)$ if $\widehat{\mu}$ is far from $\mu$. Thus, if we scan a finite cover of bounded frequencies and pick the $\widehat{\mu}$ that *minimizes* the worst–case discrepancy between the empirical characteristic function and $(1-\alpha)\, e^{2\pi i \omega \cdot \widehat{\mu}}\, \phi_D(\omega)$, the frequency witnesses force all far candidates to incur a large penalty, while all near candidates have uniformly small discrepancy. We will assume lipschitzness of $\phi_D$ which lets us discretize frequencies without losing the witness property. This is exactly what Algorithm 1 does.

We remark that, although Algorithm 1 uses a value oracle for $\phi_D$ (available for most well-known distributions) for simplicity, a sampling oracle for $D$ also suffices (see Remark B.1 for details).

We proceed with the proof of our upper bound.

**Theorem 3.2** (Upper bound via frequency witnesses)**.** *Let $\alpha \in (0, 1/2)$, $\epsilon \in (0, 1)$, and let $\mu \in \mathbb{R}^d$ be an unknown vector satisfying $\|\mu\| \leq R$, for a known $R > 1$. Fix parameters $L > 0$ and $A \in (0, 1]$, with $(1-\alpha)A - 2\alpha > 0$ and $\delta > 0$. Let $D$ be a distribution over $\mathbb{R}^d$ with density $p_D$. Assume that:*

1. *$D$ satisfies the $(\epsilon, A, \delta)$-frequency-witness condition.*
2. *$\frac{\partial}{\partial x_j} p_D \in L^1(\mathbb{R}^d)$ for all $j \in [d]$ and define $M_1 := \max_{j \in [d]} \left\|\frac{\partial}{\partial x_j} p_D\right\|_{L^1(\mathbb{R}^d)}$.*
3. *We have access to a value oracle for $\phi_D$ and $\phi_D$ is $L$-Lipschitz over $\mathcal{B}_d\left(\frac{\sqrt{d}\, M_1}{2\pi\, \delta}\right)$.*

**Algorithm 1** Robust mean estimation via frequency witnesses

---

**Input:** $\epsilon, \alpha \in (0, 1/2)$, oracle access to $\phi_D$, sample access to $D_\mu^{(\alpha)}$, the $\alpha$-mean-shift contamination of $D_\mu$ with $\|\mu\| \leq R$, parameters $R > 1$, $L, \delta > 0$ and $A, c \in (0, 1)$, and $M_1 > 0$ the derivative bound in Theorem 3.2. Assume that $D$ satisfies the $(\epsilon, A, \delta)$-frequency–witness condition and $\phi_D$ is $L$–Lipschitz.

**Output:** $\widehat{\mu}$ with $\|\widehat{\mu} - \mu\| \leq \epsilon$ w.p. $\geq 2/3$.

1. Set $B_\delta \leftarrow \frac{\sqrt{d}\, M_1}{2\pi \delta}$.

2. Set $n \leftarrow \left\lceil Cd \log\left(\frac{B_\delta RL}{\delta A}\right) \frac{1}{\left(((1-\alpha)A - 2\alpha)\delta\right)^2} \right\rceil$, for a sufficiently large constant $C > 0$.

3. Using Fact A.3, construct $\mathcal{C}_\mu$ an $\epsilon'$–cover of $\mathcal{B}_d(R)$ and $\mathcal{C}_\omega$ an $\eta$–cover of $\mathcal{B}_d(B_\delta)$, where $\epsilon' \leftarrow \min(\alpha/(2(1-\alpha)\pi B_\delta), \epsilon)$ and $\eta \leftarrow \min\{\delta/(2L), A/(2\pi R)\}$.

4. Define the set $S_\omega \leftarrow \{\omega \in \mathcal{C}_\omega : |\phi_D(\omega)| \geq \delta/2\}$.

5. Draw $x_1, \ldots, x_n \sim D_\mu^{(\alpha)}$ once; and for each $\omega \in S_\omega$ compute the empirical characteristic function of the contaminated samples, $\widehat{\phi}(\omega) = \frac{1}{n}\sum_{j=1}^n e^{2\pi i \omega \cdot x_j}$.

6. For each $\omega \in S_\omega$, set $\widehat{\psi}(\omega) \leftarrow \widehat{\phi}(\omega)/\phi_D(\omega)$.

7. For each $\widehat{\mu} \in \mathcal{C}_\mu$ and $\omega \in S_\omega$, set $\widehat{T_{\widehat{\mu}}}(\omega) \leftarrow (1-\alpha)\, e^{2\pi i\, \omega \cdot \widehat{\mu}} - \widehat{\psi}(\omega)$.

8. For each $\widehat{\mu} \in \mathcal{C}_\mu$ compute the score $s(\widehat{\mu}) \leftarrow \max_{\omega \in S_\omega} |\widehat{T_{\widehat{\mu}}}(\omega)|$.

9. Output $\widehat{\mu} \in \arg\min_{\theta \in \mathcal{C}_\mu} s(\theta)$.

---

*Then, Algorithm 1, draws*

$$n = O\left(d \log\left(\frac{\sqrt{d} M_1 RL}{\delta^2 A}\right) \frac{1}{\left(((1-\alpha)A - 2\alpha)\delta\right)^2}\right)$$

*i.i.d. $\alpha$-mean-shift contaminated samples from $D_\mu$, and outputs an estimator $\widehat{\mu}$ satisfying $\|\widehat{\mu} - \mu\| \leq \epsilon$ with probability at least $2/3$.*

Before we proceed with the proof, we briefly comment on the conditions. First, the bounded-mean assumption is easy to reduce to algorithmically for many distributions. For instance, when the distribution has bounded covariance, one can use adversarially robust estimators and then center the data to obtain a bounded mean (see Fact A.4). Second, the Lipschitz continuity of $\phi_D$ is implied by finite absolute first

moment of $D$. Indeed, by the definition of the characteristic function and standard inequalities for the exponential function, one obtains the required Lipschitz bound.

*Proof of Theorem 3.2.* First, we show that every $(\epsilon, A, \delta)$ frequency witness has bounded norm; this will help us bound the frequency cover radius. The proof uses the fact that smoothness of a function results in faster decay of its Fourier transform (see Appendix B for the proof).

*Claim* 3.3 (Every frequency witness has bounded norm). Fix $\epsilon, A, \delta \in (0, 1)$. If $\omega$ is an $(\epsilon, A, \delta)$-frequency-witness for some direction $v$ (i.e., $|\sin(\pi v \cdot \omega)| \geq A$ and $|\phi_D(\omega)| \geq \delta$), then necessarily $\|\omega\| \leq B_\delta$ where $B_\delta := \frac{\sqrt{d}\, M_1}{2\pi \delta}$.

Now note that since any frequency witness $\omega$ necessarily satisfies $\|\omega\| \leq B_\delta$ (by Claim 3.3), it suffices to construct the frequency cover inside $\mathcal{B}_d(B_\delta)$.

Let $\mathcal{C}_\mu$ be the $\epsilon'$-cover of $\mathcal{B}_d(R)$ with $\epsilon' = \min(\alpha/(2(1-\alpha)\pi B_\delta), \epsilon)$ and $\mathcal{C}_\omega$ be the $\eta$-cover of $\mathcal{B}_d(B_\delta)$ with $\eta = \min(\delta/(2L), A/(2\pi R))$, constructed in Line 3. By Fact A.3 we have that $|\mathcal{C}_\mu| \leq (5R/\epsilon')^d$ and $|\mathcal{C}_\omega| \leq (5B_\delta/\eta)^d$. Let $\widehat{\phi} : \mathbb{R}^d \to \mathbb{C}$ be the empirical characteristic function of the contaminated samples and by $\phi$ its population counterpart. For the set of $\omega \in \mathbb{R}^d$ where $\phi_D(\omega) \neq 0$ define the function $\widehat{\psi}(\omega) := \widehat{\phi}(\omega)/\phi_D(\omega)$. Note that $\psi(\omega) := \mathbf{E}[\widehat{\psi}(\omega)] = (1-\alpha)e^{2\pi i \omega \cdot \mu} + \alpha\phi_Q(\omega)$. Also denote by $\widehat{T_{\widehat{\mu}}}(\omega) := (1-\alpha)e^{2\pi i \omega \cdot \widehat{\mu}} - \widehat{\psi}(\omega)$ the test function computed in Line 7 and by $T_{\widehat{\mu}}(\omega) := \mathbf{E}[\widehat{T_{\widehat{\mu}}}(\omega)] = (1-\alpha)e^{2\pi i \omega \cdot \widehat{\mu}} - \psi(\omega)$ its population counterpart. Denote the set $\Omega(\delta, B_\delta) := \{\omega \in \mathbb{R}^d : |\phi_D(\omega)| > \delta, \|\omega\| \leq B_\delta\}$. Note that Algorithm 1 essentially consists of finding a $\widehat{\mu} \in \mathcal{C}_\mu$ such that $\max_{\omega \in \Omega(\delta, B_\delta)} T_{\widehat{\mu}}(\omega)$ is small.

First we prove that for a candidate mean, $\widehat{\mu}$, if the direction $\widehat{\mu} - \mu$ exhibits a frequency-witness $\omega$, then $T_{\widehat{\mu}}(\omega)$ is large. For the proof, we refer the reader to Appendix B.

*Claim* 3.4 (Distant candidates have a large $T_{\widehat{\mu}}$). Let $\widehat{\mu} \in \mathbb{R}^d$ be candidate mean such that $\|\widehat{\mu} - \mu\| > \epsilon$. If $\omega \in \mathbb{R}^d$ is a $(\epsilon, A, \delta)$-frequency-witness for the direction $v := \widehat{\mu} - \mu$, then $|T_{\widehat{\mu}}(\omega)| \geq 2(1-\alpha)A - \alpha$.

Now we prove that if a candidate mean $\widehat{\mu}$ is sufficiently close, then $T_{\widehat{\mu}}(\omega)$ is small for all $\omega$ in a large ball. For the proof, we refer the reader to Appendix B.

*Claim* 3.5 (Close candidates have small $T_{\widehat{\mu}}$). Let $\widehat{\mu} \in \mathbb{R}^d$ be a candidate mean and set $v := \widehat{\mu} - \mu$. Then, $|T_{\widehat{\mu}}(\omega)| \leq 2(1-\alpha)\pi\|\omega\|\|v\| + \alpha$.

The next claim shows that if you already have a good frequency witness for a direction $v$, then passing to an $\eta$-cover does not destroy it. Instead, you can find a nearby point in the cover that remains a frequency witness, with only small

losses in the parameters. For the proof, we refer the reader to Appendix B.

*Claim* 3.6 (Covering over $\omega$ preserves frequency witnesses). Fix $v \in \mathbb{R}^d$. Let $\mathcal{C}_\omega$ be an $\eta$-cover of $\mathcal{B}_d(B_\delta)$. If there exists a $(\epsilon, A, \delta)$-frequency-witness for the direction $v$, then there exists an $\omega \in \mathcal{C}_\omega$ that is a $(\epsilon, A - \pi\eta \|v\|, \delta - \eta L)$-frequency-witness for $v$.

We now prove that we can compute $\widehat{T}_\mu(\omega)$ efficiently with samples for all $\widehat{\mu} \in \mathcal{C}_\mu, \omega \in \mathcal{C}_\omega$. This essentially follows from the fact that the characteristic function is the average of unit absolute value complex numbers (for the proof we refer the reader to Appendix B).

*Claim* 3.7 (Concentration of $\widehat{T}$). Fix a sufficiently large universal constant $C > 0$ and $\eta, \tau \in (0, 1)$. If $n \geq C \log(|\mathcal{C}_\omega|/\tau)/(\eta\delta)^2$, then with probability at least $1 - \tau$ for any $\omega \in \mathcal{C}_\omega$ such that $|\phi_D(\omega)| \geq \delta$ and any $\widehat{\mu} \in C_\mu$ it holds that $|T_{\widehat{\mu}}(\omega) - \widehat{T}_{\widehat{\mu}}(\omega)| \leq \eta$.

Finally we can proceed with the proof of the main theorem.

Denote by $S_\omega$ the search set of frequencies used by the algorithm $S_\omega = \{\omega \in \mathcal{C}_\omega : \phi_D(\omega) > \delta\}$ (see Line 4). Note that since $\mathcal{C}_\omega$ is an $\eta$-cover of $\mathcal{B}_d(B_\delta)$ with $\eta = \min(\delta/(2L), A/(2\pi R))$, by Claim 3.6 we have that for every $v \in \mathcal{B}_d(R)$ there exist an $(\epsilon, A/2, \delta/2)$-frequency-witness in $\mathcal{C}_\omega$. Hence from Claim 3.4 we have that for any $\widehat{\mu}$ such that $\|\widehat{\mu} - \mu\| \geq \epsilon$ there exists an $\omega \in S_\omega$ such that $T_{\widehat{\mu}}(\omega) > T_{\text{bad}} := (1 - \alpha)A - \alpha$. While from Claim 3.5 we have that for a $\widehat{\mu}$ such that $\|\widehat{\mu} - \mu\| \leq \epsilon'$ (note that $\epsilon' \leq \epsilon$) for any $\omega \in S_\omega$, $T_{\widehat{\mu}}(\omega) < T_{\text{good}} := 2(1 - \alpha)\pi B_\delta \epsilon' + \alpha$. Denote for convenience $c = (1 - \alpha)A/2 - \alpha$. Note that since $\epsilon' < ((1 - \alpha)A - 2\alpha)/(4(1 - \alpha)\pi B_\delta)$, we have that $T_{\text{bad}} > T_{\text{good}} + c$. Therefore, by Claim 3.7 and the fact that $n \geq Cd \log(B_\delta/\eta)/(c\delta)^2$ for a sufficiently large constant $C$ we have that $\max_{\omega \in S_\omega} \widehat{T}_{\widehat{\mu}'}(\omega) < \max_{\omega \in S_\omega} \widehat{T}_{\widehat{\mu}}(\omega)$ for any $\mu, \mu' \in \mathcal{C}_\mu$ such that $\|\widehat{\mu} - \mu\| \geq \epsilon$ and $\|\widehat{\mu}' - \mu\| \leq \epsilon'$. Which implies that no $\widehat{\mu}$ with $\|\widehat{\mu} - \mu\| \geq \epsilon$ can be returned by the algorithm since there exists a candidate $\widehat{\mu}' \in \mathcal{C}_\mu$ with $\|\widehat{\mu}' - \mu\| \leq \epsilon'$ and hence with $\max_{\omega \in S_\omega} \widehat{T}_{\widehat{\mu}'}(\omega) < \max_{\omega \in S_\omega} \widehat{T}_{\widehat{\mu}}(\omega)$. $\square$

## 4. Sample Complexity Lower Bound

In this section, we derive a lower bound for one-dimensional distributions under the negation of the assumption used for the algorithm together with some other mild tail bounds on the distributions.

Recall that the Frequency-witness condition (Definition 3.1) used for the upper bound states that for all directions $v \in \mathbb{R}^d$ with $\|v\| \geq \epsilon$ there exists a frequency $\omega \in \mathbb{R}^d$ such that $v \cdot \omega$ is far from an integer and $|\phi_D(\omega)|$ is bounded away from 0. Our 1d hard instances satisfy (a form of) the negation of

this condition: $\phi_D(\omega)$ has negligible $L^2$ mass whenever $\epsilon\omega$ is far from an integer, i.e., its Fourier mass concentrates in thin bands around the lattice $\{\omega \approx k/\epsilon : k \in \mathbb{Z}\}$.

Additionally, we impose an assumption controlling the tails of our distribution. This assumption ensures that the distribution has sufficient tail decay to relate the total variation distance (the $L^1$ norm) to the $L^2$ norm, and hence to convert Fourier-transform closeness into $L^2$ closeness via Parseval's identity. Moreover, for our applications, the parameter for quantifying this decay is constant and therefore does not significantly affect our lower-bound complexity.

In particular, we present the following theorem:

**Theorem 4.1** (1-Dimensional Lower Bound). *There exists a sufficiently large universal constant $C > 0$ such that the following hold. Fix $\alpha \in (0, 1/2)$ and $\epsilon, \delta, \sigma > 0$ with $\sigma > C\delta$. Let $D$ be a distribution over $\mathbb{R}$ that satisfies the following conditions:*

1. *$\|\phi_D(\omega)\mathbb{1}\{\text{dist}(\epsilon\omega, \mathbb{Z}) > c\alpha\}\|_{L^2} \leq \delta$, for a sufficiently small constant $c > 0$.*
2. *For all $R > 0$, $\mathbf{Pr}_{x \sim D}[|x| \geq R] \leq \sigma/R$.*

*Then, any algorithm that estimates the mean to error at most $\epsilon$, with probability at least $2/3$, from $\alpha$-mean-shift contaminated samples (as in Definition 1.1), must use*

$$\Omega\left(\frac{1}{(\delta\sigma)^{1/3} + (\epsilon/\alpha)^3(\delta/\sigma)^2}\right)$$

*samples. In particular, if $(\epsilon/\alpha)^3(\delta/\sigma)^2 \leq (\delta\sigma)^{1/3}$, then this simplifies to $\Omega(1/(\delta\sigma)^{1/3})$ samples.*

Before we move to the proof we make some important remarks about the statement of Theorem 4.1.

*Remark* 4.2 (High-Dimensional Lower Bound). While Theorem 4.1 is a lower bound against univariate distributions, it also induces a lower bound against multivariate distributions by considering product distributions. In particular, let $D$ be a product distribution over $\mathbb{R}^d$ whose coordinate marginals are independent and each satisfy the conditions of Theorem 4.1. In all but rare cases, estimating the mean to error $\epsilon$ requires at least $\Omega(d)$ samples. Moreover, since the coordinates are independent, the one-dimensional lower bound (denoted by lb) still applies: we may consider an instance in which both the mean displacement and the noise lie entirely in the first coordinate. As a result, $\max(\text{lb}, \Omega(d)) \geq (\text{lb} + \Omega(d))/2$ is a lower bound. By Hölder's inequality, this implies a lower bound of $(\Omega(d))^{1/p}\text{lb}^{1/q}$ for $1/p + 1/q = 1$. Concretely, by taking $p$ close to 1, this yields a lower bound of $d^{0.99}e^{\Omega((\alpha/\epsilon)^2)}$ against $d$-dimensional Gaussian distributions, which is tight up to polynomial factors.

*Remark* 4.3 (Condition of Upper & Lower Bound). We remark that the condition $\sin(\pi\omega \cdot v) \geq A$ is equivalent to requiring that $v \cdot \omega$ be bounded away from integers: indeed

$|\sin(\pi t)| = |\sin(\pi \operatorname{dist}(t, \mathbb{Z}))|$, so $|\sin(\pi \omega \cdot v)| \geq A$ iff $\operatorname{dist}(v \cdot \omega, \mathbb{Z}) \geq \arcsin(A)/\pi$. Thus, the frequency–witness condition demands frequencies $\omega$ where $\omega \cdot v$ is far from an integer and $D$ has nontrivial Fourier mass. In contrast, the lower bound's assumes that the $L^2$–mass of $\phi_D$ concentrates within small bands around integer points $\{\omega : \operatorname{dist}(v \cdot \omega, \mathbb{Z}) \leq c\alpha\}$, i.e., regions where $|\sin(\pi \omega \cdot v)| \leq \sin(\pi c\alpha)$ is small. Therefore, the regions that both conditions consider are essentially the same.

Now, the only mismatch is that the witness condition uses the $L^\infty$ norm of $\phi_D$ over $S := \omega \in \mathbb{R} : \operatorname{dist}(\epsilon\omega, \mathbb{Z}) > c\alpha$, whereas the lower-bound condition uses the $L^2$ norm. However, under an additional derivative assumption (which holds for all the examples studied in Appendix D), Claim C.1 implies that $|\phi_D \mathbb{1}_S|_{L^2} \lesssim \sqrt{|\phi_D \mathbb{1}_S|_{L^\infty}}$. Therefore, the upper and lower-bound conditions relate polynomially to each other. More generally, stronger smoothness yields faster Fourier decay and an even tighter link between pointwise and $L^2$ control.

*Remark* 4.4 (Tail Condition). Condition (2) of Theorem 4.1 is a mild tail-decay requirement: it bounds the mass of $D$ outside $[-R, R]$ by $\sigma/R$, which is exactly what we need to control the truncation error in Lemma 4.5. In particular, if $\mathbf{E}_D[|x|] \leq \sigma$, then $\mathbf{Pr}_D(|x| \geq R) \leq \sigma/R$ for all $R > 0$ by Markov's inequality. Bounded variance also implies Condition (2) as it implies bounded $\mathbf{E}_D[|x|]$.

To derive our lower bound, we construct two distributions $P$ and $Q$ whose characteristic functions are nearly indistinguishable on a large set of frequencies (the complement of the set in condition (1) of Theorem 4.1). Let $p, q$ be the density functions of $P, Q$. To convert this Fourier-space closeness into a statistical complexity lower bound, we need an inequality that controls the distributional distance between distributions by the Fourier discrepancies of their characteristic functions. The next lemma provides the key bridge: the total variation distance $d_{\mathrm{TV}}(P, Q)$ can be bounded in terms of the $L^2$-norm of the characteristic-function difference, $\|\Delta\phi\|_{L^2}$. and the $L^1$ discrepancy between the two densities on the tails, $\|(p-q)\mathbb{1}\{|x| > R\}\|_{L^1}$. The proof is fairly standard and is deferred to Appendix C.

**Lemma 4.5** (Characteristic function to TV distance closeness). *Let $P$ and $Q$ be distributions over $\mathbb{R}$ with densities $p$ and $q$ respectively. Denote by $\Delta\phi := \phi_P - \phi_Q$ the difference of their characteristic functions. Then, for every $R > 0$,*

$$d_{\mathrm{TV}}(P, Q) \leq \frac{1}{2}\|(p-q)\mathbb{1}\{|x| > R\}\|_{L^1} + \sqrt{\frac{R}{2}}\,\|\Delta\phi\|_{L^2}\,.$$

Next we present the Fourier construction that we will use for our lower bound. By Lemma 4.5, it is enough to control $\|\Delta\phi\|_{L^2}$ and $\|(p - q)\mathbb{1}\{|x| > R\}\|_{L^1}$. The hypothesis on $D$ controls the $L^2$ contribution away from the lattice bands, since on that region $\phi_D$ is small in $L^2$. Thus, the

main difficulty is controlling $\Delta\phi$ on the bands where $|\phi_D|$ may be large. Fix $A > 0$ (to be chosen) and suppose that $|\epsilon\omega - k| \leq A$ for some $k \in \mathbb{Z}$. Note that, without noise, we have

$$
\begin{aligned}
|\phi_{D_{\epsilon/2}}(\omega) - \phi_{D_{-\epsilon/2}}(\omega)| &= |\phi_D(\omega)(e^{\pi i \epsilon \omega} - e^{-\pi i \epsilon \omega})| \\
&= 2|\phi_D(\omega)\sin(\pi\epsilon\omega)| \\
&\leq 2|\phi_D(\omega)|\,|\sin(\pi A)|\,.
\end{aligned}
$$

While the factor $|\sin(\pi A)|$ is small for small $A$, this does not by itself control the $L^2$ norm on the union of bands, because $\phi_D$ is otherwise unconstrained there and may have large (and even infinite) $L^2$ mass (e.g., for $D = \mathcal{N}(0, \sigma^2)$, $\phi_D(\omega) = e^{-2\pi^2\sigma^2\omega^2}$ and $\|\phi_D(\omega)\|_{L^2}^2 = \sqrt{\pi}/(2\pi\sigma)$, which can be large for small $\sigma$). However, for any point $\omega$ in this region the amplitude bound $|\phi_D(\omega)|\,|\sin(\pi A)| \leq |\sin(\pi A)|$ is small when $A$ is small; in our setting we will take $A$ on the order of the noise rate $\alpha$.

This makes it possible to design two noise distributions such that the difference of their characteristic functions, multiplied by the noise rate $\alpha$, approximates $(e^{\pi i \epsilon \omega} - e^{-\pi i \epsilon \omega})$ arbitrarily well on this region, essentially canceling its contribution to $d_{\mathrm{TV}}$. Crucially, this matching is performed via a smooth periodic window, which also yields the tail control needed in Lemma 4.5. Together, these two properties yield the desired result.

The next lemma formalizes this Fourier matching construction (see Appendix C for the proof).

**Lemma 4.6** (Fourier Matching). *Let $\epsilon, \alpha \in (0, 1)$, $\widehat{f}(\omega) := (1 - \alpha)(e^{\pi i \epsilon \omega} - e^{-\pi i \epsilon \omega})/\alpha$ and $f := \mathcal{F}^{-1}[\widehat{f}]$. For every $w > 0$ there exists a function $\widehat{\rho}_w : \mathbb{R} \to \mathbb{C}$ with $\widehat{\rho}_w(\omega) = 1$ for all $\omega : |\omega - i/\epsilon| \leq w$ for some $i \in \mathbb{Z}$, $\widehat{g} := \widehat{f} \cdot \widehat{\rho}_w$, and $g := \mathcal{F}^{-1}[\widehat{g}]$, such that the following hold:*

*1. $g$ is a real valued signed measure.*
*2. $\int_{-\infty}^{\infty} g(x)\, dx = 0$.*
*3. $\|g(x)\|_{L^1} \lesssim \epsilon w/\alpha$.*
*4. For every $R > \max\{2\epsilon, 2/w\}$ it holds that $\|g\mathbb{1}\{|x| > R\}\|_{L^1} \lesssim \frac{\epsilon}{\alpha}\frac{1}{w^2 R^3}$.*

We remark on the difference of the above construction with previous work:

*Remark* 4.7 (Construction of (Kotekal & Gao, 2025)). Our Fourier matching differs from (Kotekal & Gao, 2025). Their work focuses only on the Gaussian setting and enforces $\widehat{P}_0(\omega) = \widehat{P}_1(\omega)$ on an interval around the origin. For general distributions, matching only around $\omega = 0$ is not sufficient. Instead, we apply a smooth window that equals 1 on $[-w, w]$ and vanishes outside $[-2w, 2w]$, and then periodize over multiples of $1/\epsilon$. In this way, the discrepancy is canceled across the entire set $\omega : \operatorname{dist}(\omega\epsilon, \mathbb{Z}) \leq c\alpha$. Moreover, the smoothness of the window ensures that our construction is realized by distributions and provides the tail control needed for our total-variation bounds.

## 4.1. Proof of Theorem 4.1

In this section, we prove our main lower bound result.
*Proof of Theorem 4.1.* Let $Q_0, Q_1$ be arbitrary distributions over $\mathbb{R}$. We define the distributions $P_0 := (1 - \alpha)\delta_{\epsilon/2} * D + \alpha Q_0 * D$ and $P_1 := (1 - \alpha)\delta_{-\epsilon/2} * D + \alpha Q_1 * D$. Note that

$$
\begin{aligned}
\Delta\phi(\omega) &:= \phi_{P_0}(\omega) - \phi_{P_1}(\omega) \\
&= \phi_D(\omega)\Big((1 - \alpha)(e^{\pi i \epsilon \omega} - e^{-\pi i \epsilon \omega}) \\
&\qquad + \alpha(\phi_{Q_0}(\omega) - \phi_{Q_1}(\omega))\Big).
\end{aligned}
$$

Note that any finite signed measure $h$ on $\mathbb{R}$ with $\|h\|_{L^1} = 2$ and $\int_{\mathbb{R}} h(x)dx = 0$ can be realized as the difference of two probability distributions by setting $Q_0$ to be $(h(x))_+$ and $Q_1$ to be $(h(x))_-$ (where $h_+$ and $h_-$ the Jordan decomposition of $h$). Therefore, by applying Lemma 4.6 with $w = c\alpha/\epsilon$ for a sufficiently small constant $c > 0$, we get the existence of a signed measure $g$ with $\|g\|_{L^1} \le 2$. Let $m := \|g\|_{L^1}/2 < 1$ and define $Q_0$ and $Q_1$ as the probability measures

$$
Q_0 := g_- + (1 - m)\delta_0, \qquad Q_1 := g_+ + (1 - m)\delta_0,
$$

respectively. Note that $Q_0 - Q_1 = -g$. Therefore, we obtain the existence of $Q_0, Q_1$ such that $\Delta\phi(\omega) = 0$ for all $\omega \in \mathbb{R}$ satisfying $|\omega - i/\epsilon| \le c\alpha/\epsilon$ for some $i \in \mathbb{Z}$.

Now we use this fact along with our assumptions to bound the total variation distance between $P_0$ and $P_1$. For notational simplicity let $E_0 := (1 - \alpha)\delta_{\epsilon/2} + \alpha Q_0$, $E_1 := (1 - \alpha)\delta_{-\epsilon/2} + \alpha Q_1$ and denote by $\Delta\phi_E(\omega) = \phi_{E_0}(\omega) - \phi_{E_1}(\omega)$. Thus, we have that $\Delta\phi = \phi_D\Delta\phi_E$ and $P_j = D * E_j$.

We will use Lemma 4.5 to bound $d_{\mathrm{TV}}(P_0, P_1)$. Let $p_0, p_1$ be the densities of $P_0, P_1$, and write $h := p_0 - p_1$. By Lemma 4.5, for every $R > 0$,

$$
d_{\mathrm{TV}}(P_0, P_1) \le \frac{1}{2}\|h(x)\mathbb{1}\{|x| > R\}\|_{L^1} + \sqrt{\frac{R}{2}}\|\Delta\phi\|_{L^2}.
$$

We first bound the Fourier term. Since $|\Delta\phi_E(\omega)| \le 2$ by Fact A.1, we have

$$
\begin{aligned}
\|\Delta\phi\|_{L^2} &= \|\phi_D(\omega)\Delta\phi_E(\omega)\mathbb{1}\{\mathrm{dist}(\epsilon\omega, \mathbb{Z}) > c\alpha\}\|_{L^2} \\
&\le 2\,\|\phi_D(\omega)\mathbb{1}\{\mathrm{dist}(\epsilon\omega, \mathbb{Z}) > c\alpha\}\|_{L^2} \le 2\delta,
\end{aligned}
$$

where in the last inequality we used the assumption that $\|\phi_D(\omega)\mathbb{1}\{\mathrm{dist}(\epsilon\omega, \mathbb{Z}) > c\alpha\}\|_{L^2} \le \delta$. Therefore,

$$
(R/2)^{1/2}\|\Delta\phi\|_{L^2} \lesssim \sqrt{R}\,\delta.
$$

It remains to bound the tail term $\|h\mathbb{1}\{|x| > R\}\|_{L^1}$. Since $P_j = D * E_j$, we have $h = p_0 - p_1 = D * (E_0 - E_1)$.

Let $\Delta E := E_0 - E_1$ (a finite signed measure). Using $|D * \Delta E| \le D * |\Delta E|$, we get

$$
\begin{aligned}
&\|h\mathbb{1}\{|x| > R\}\|_{L^1} \\
&= \int_{|x|>R} \big|(D * \Delta E)(x)\big|\,dx \le \int_{|x|>R} (D * |\Delta E|)(x)\,dx.
\end{aligned}
$$

Define $E'$ as the measure $|\Delta E(x)|/\int_{\mathbb{R}}|\Delta E|(y)dy$. Now by the union bound we have multiplying and dividing by $\int_{\mathbb{R}}|\Delta E|(y)dy$ gives us

$$
\begin{aligned}
&\int_{|x|>R} (D * |\Delta E|)(x)\,dx \\
&\le \mathbf{Pr}_D[|x| > R/2]\int_{\mathbb{R}}|\Delta E|\mathrm{d}y + \int_{|y|>R/2}|\Delta E|\mathrm{d}y\,.
\end{aligned}
$$

Since $E_0, E_1$ are probability distributions, $\int_{\mathbb{R}}|\Delta E|\mathrm{d}y \le 2$. Therefore,

$$
\|h\mathbb{1}\{|x|>R\}\|_{L^1} \le 2\mathbf{Pr}_D[|x| > R/2] + \int_{|y|>R/2}|\Delta E|\mathrm{d}y.
$$

Next we bound $\int_{|y|>R/2}|\Delta E|\mathrm{d}y$ using the structure of $\Delta E$. By definition, $\Delta E = (1 - \alpha)(\delta_{\epsilon/2} - \delta_{-\epsilon/2}) + \alpha(Q_0 - Q_1)$. By our choice of $Q_0, Q_1$ we have $\Delta E = (1 - \alpha)(\delta_{\epsilon/2} - \delta_{-\epsilon/2}) - \alpha g$. Thus $|\Delta E| \le (1 - \alpha)(\delta_{\epsilon/2} + \delta_{-\epsilon/2}) + \alpha|g|$. In particular, for $R > \epsilon$

$$
\begin{aligned}
\int_{|y|>R/2}|\Delta E|\mathrm{d}y &\le \alpha\int_{|y|>R/2}|g(y)|\,dy \\
&= \alpha\|g\mathbb{1}\{|y| > R/2\}\|_{L^1}.
\end{aligned}
$$

Consider $R$ greater than a sufficiently large constant. By item (4) of Lemma 4.6 with radius $R/2$ (note that $R > \max(4\epsilon, 4/w)$, since we have that $1/w = O(\epsilon/\alpha) = O(1)$ and $R$ is greater than a sufficiently large constant), we obtain

$$
\|g\mathbb{1}\{|y| > R/2\}\|_{L^1} \lesssim \frac{\epsilon}{\alpha} \cdot \frac{1}{w^2(R/2)^3} \lesssim \left(\frac{\epsilon}{\alpha R}\right)^3.
$$

Combining the above bounds gives

$$
\|h\mathbb{1}\{|x| > R\}\|_{L^1} \le 2\mathbf{Pr}_{x\sim D}[|x|>R/2] + O\left(\left(\frac{\epsilon}{\alpha R}\right)^3\right).
$$

Therefore,

$$
d_{\mathrm{TV}}(P_0, P_1) \lesssim \frac{\sigma}{R} + \left(\frac{\epsilon}{\alpha R}\right)^3 + \sqrt{R}\,\delta.
$$

Choosing $R = (\sigma/\delta)^{2/3}$, note that $\sigma/\delta$ is assumed to be less than a sufficiently large constant, we have

$$
d_{\mathrm{TV}}(P_0, P_1) \lesssim (\delta\sigma)^{1/3} + (\epsilon/\alpha)^3(\delta/\sigma)^2.
$$

Finally, consider the decision problem of having samples generated from either $P_0$ or $P_1$ and choosing which generated the samples. This problem is harder than estimating the

mean under the mean-shift-contamination model to error $\epsilon/2$, since then means associated with $P_0, P_1$ have distance $\epsilon$. Given that $P_0, P_1$ are drawn with probability $1/2$ to generate the samples, then using $n$ i.i.d. samples we have that the optimal error for the decision problem is

$$p^\star(n) = 1/2\left(1 - d_{TV}(P_0^{\otimes n}, P_1^{\otimes n})\right) .$$

By tensorization, $d_{TV}(P_0^{\otimes n}, P_1^{\otimes n}) \leq 1 - (1 - d_{TV}(P_0, P_1))^n$, hence $p^\star(n) \geq 1/2(1 - d_{TV}(P_0, P_1))^n$. Hence for achieving error less than $1/3$, it is necessary that $(1 - d_{TV}(P_0, P_1))^n \leq 2/3$ which implies that $n \geq \log(3/2)/d_{TV}(P_0, P_1) = \Omega(1/d_{TV}(P_0, P_1))$. Combining with the bound above concludes the proof of Theorem 4.1. $\qquad\square$

## Acknowledgments

ID was supported by NSF Medium Award CCF-2107079, ONR award number N00014-25-1-2268 and an H.I. Romnes Faculty Fellowship. GI was supported by ONR award number N00014-25-1-2268. DK was supported by NSF Medium Award CCF-2107547.

## Impact Statement

The primary contribution of this research lies in theoretical advancements and foundational insights. Therefore, we believe that it does not have direct implications for societal, ethical, or policy-related issues.

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

# Appendix

The appendix is structured as follows: First Appendix A includes additional preliminaries required in subsequent technical sections. Appendix C contains the proofs of the technical lemmas of the lower bound (Section 4). Finally, Appendix D contains applications of our theorems to well-known distributions.

## A. Omitted Facts and Preliminaries

**Additional Notation:** We define the Dirac delta function $\delta(\cdot)$ as the distribution with the property that for any measurable $f : \mathbb{R}^d \to \mathbb{R}$, we have $\int_{\mathbb{R}^d} f(x)\delta(x)dx = f(0)$. We denote by $\chi_w : \mathbb{R} \to \mathbb{R}$ the rectangular window function $\chi_w(x) := \mathbb{1}\{x \in [-w, w]\}$. For a complex number $z \in \mathbb{C}$, $z = a + bi$ we denote by $\overline{z}$ its conjugate $\overline{z} = a - bi$. We extend the $L^1$-notation to finite signed measures (and, more generally, distributions) by duality: for such $g$ define $\|g\|_{L^1} := \sup_{\|f\|_\infty \le 1} \langle g, f \rangle$, where $\langle g, f \rangle = \int f \, dg$ when $g$ is a (signed) measure, so in particular $\|\delta_z\|_{L^1} = 1$; when $g$ admits a density $g(x) \, dx$ this agrees with $\int_{\mathbb{R}^d} |g(x)| \, dx$. Let $f^{*m}$ denote the $m$-fold convolution of $f$ with itself, i.e., $f^{*m} = f * \cdots * f$ with $m$ factors.

*Fact* A.1 (Useful Fourier Transform facts (see (Stein & Shakarchi, 2011))). Let $f, \phi$ be finite measures and let $\phi(\omega) = \mathcal{F}[f]$. Then the following properties hold:

(i) If $\phi$ is a function and $\phi(-\omega) = \overline{\phi(\omega)}$ then for all $\omega \in \mathbb{R}$, then $f$ is a real-valued measure.

(ii) If $\int_{\mathbb{R}} |f(x)|dx \le 1$, then $|\phi(\omega)| \le 1$, for all $\omega \in \mathbb{R}$.

(iii) If $\phi$ is $T$-periodic for some $T > 0$, then the inverse Fourier transform is the discrete measure

$$f(x) = \sum_{k \in \mathbb{Z}} a_k \, \delta(x - k/T) \text{ with } a_k = \frac{1}{T} \int_0^T \phi(\omega)e^{-2\pi i k\omega/T} \, d\omega, k \in \mathbb{Z} \,.$$

*Fact* A.2 (Fourier Derivatives (Grafakos et al., 2008)). Let $f : \mathbb{R}^d \to \mathbb{R}$ be absolutely integrable function with absolutely integrable partial derivatives and denote $\widehat{f} := \mathcal{F}[f]$. Then, for all $j \in [d]$ it holds that $\mathcal{F}[\partial f / \partial x_j] = -2\pi i \omega_j \widehat{f}(\omega)$

*Fact* A.3 ($\epsilon$-Cover of a Ball (see (Vershynin, 2010))). Let $R > 0$ and $d \in \mathbb{Z}_+$. There exists an $\epsilon$-cover $G$ of $\mathcal{B}_d(R)$ in $\ell_2$ such that $\log |G| \le C \, d \log(R/\epsilon)$, for a universal constant $C > 0$. In particular, one may take $|G| \le (1 + 4R/\epsilon)^d$.

*Fact* A.4 (Adversarially Robust Estimator (Diakonikolas & Kane, 2023)). Let $0 < \alpha < 1/3$ and let $D$ be a distribution over $\mathbb{R}^d$ with $\mu := \mathbf{E}_{x \sim D}[x]$ such that $\text{Cov}(D) \preceq \sigma^2 I$. There exists an algorithm that, given $\alpha$, $\sigma$, and $N + O(d \log(1/\delta)/\alpha)$ i.i.d. Huber $\alpha$-corrupted samples (samples from a distribution of the form $(1 - \alpha)D + \alpha N$ for an arbitrary distribution $N$ over $\mathbb{R}^d$), in $\text{poly}(Nd)$ time returns $\widehat{\mu}$ such that with probability $1 - \delta$ it holds $\|\widehat{\mu} - \mu\| = O(\sigma\sqrt{\alpha})$.

## B. Omitted Content from Section 3

In this section, we provide the content omitted from our upper bound section.

*Remark* B.1 (Access to distribution $D$). We remark that even though Algorithm 1 uses a value oracle for $\phi_D$ (which is available for most well-known distributions) for simplicity, a sampling oracle for $D$ suffices. In particular, because $|\phi_D(\omega)| \le 1$ and all evaluations occur on a finite grid $\mathcal{C}_\omega$, the empirical characteristic function $\widehat{\phi}_D$ concentrates uniformly on $\mathcal{C}_\omega$ with a modest number of clean samples. Hence both uses of $\phi_D$ can be handled from data: (i) to construct $S_\omega$, we can threshold $\widehat{\phi}_D$ at a slightly smaller level so that frequency witnesses are retained and every included $\omega$ has $|\phi_D(\omega)|$ bounded away from 0 w.h.p.; and (ii) to compute $\widehat{\psi}$, we can divide by $\widehat{\phi}_D(\omega)$ instead of $\phi_D(\omega)$, and $\widehat{\psi}$ turns out to be stable on the new $S_\omega$ since $|\phi_D(\omega)|$ bounded away from 0. Consequently, with an additional clean sample budget on the order of the contaminated budget, a sampling oracle fully replaces the value oracle without altering the stated rates.

In what follows, we provide the proofs of the claims in Theorem 3.2. We remind the reader that $B_\delta, \phi, \widehat{\phi}, T_{\widehat{\mu}}, \widehat{T}_{\widehat{\mu}}, \psi, \widehat{\psi}, D, Q, \phi_D$ and $\phi_Q$ are the quantities defined in the proof of Theorem 3.2.

*Claim* B.2 (Every frequency witness has bounded norm). Fix $\epsilon, A, \delta \in (0, 1)$. If $\omega$ is an $(\epsilon, A, \delta)$-frequency-witness for some direction $v$ (i.e., $|\sin(\pi v \cdot \omega)| \ge A$ and $|\phi_D(\omega)| \ge \delta$), then necessarily $\|\omega\| \le B_\delta$ where $B_\delta := \frac{\sqrt{d} \, M_1}{2\pi \, \delta}$ .

*Proof of Claim B.2.* Since $A > 0$, any frequency witness must satisfy $\omega \ne 0$.

Let $\omega \in \mathbb{R}^d$ and choose an index $j \in [d]$. By Fact A.2 we have

$$\int_{\mathbb{R}^d} \frac{\partial}{\partial x_j} p_D(x)\, e^{2\pi i\, \omega \cdot x}\, dx = 2\pi i \omega_j \phi_D(\omega)\ .$$

Which implies

$$\phi_D(\omega) = \frac{1}{2\pi i\, \omega_j} \int_{\mathbb{R}^d} \frac{\partial}{\partial x_j} p_D(x)\, e^{2\pi i\, \omega \cdot x}\, dx\ .$$

By choosing $j$ such that $\|\omega_j\| = \|\omega\|_\infty$ we have

$$|\phi_D(\omega)| \le \frac{1}{2\pi |\omega_j|} \left\| \frac{\partial}{\partial x_j} p_D \right\|_{L^1(\mathbb{R}^d)} \le \frac{M_1}{2\pi \|\omega\|_\infty} \le \frac{\sqrt{d}\, M_1}{2\pi \|\omega\|}\ .$$

Which concludes the proof of Claim B.2. $\qquad\square$

*Claim* B.3 (Distant candidates have a large $T_{\widehat{\mu}}$). Let $\widehat{\mu} \in \mathbb{R}^d$ be candidate mean such that $\|\widehat{\mu} - \mu\| > \epsilon$. If $\omega \in \mathbb{R}^d$ is a $(\epsilon, A, \delta)$-frequency-witness for the direction $v := \widehat{\mu} - \mu$, then $|T_{\widehat{\mu}}(\omega)| \ge 2(1 - \alpha)A - \alpha$.

*Proof.* Note that

$$T_{\widehat{\mu}}(\omega) = e^{2\pi i \omega \cdot (\mu + v/2)}(1 - \alpha)(e^{\pi i \omega \cdot v} - e^{-\pi i \omega \cdot v}) - \alpha \phi_Q(\omega)$$
$$= e^{2\pi i \omega \cdot (\mu + v/2)} 2i(1 - \alpha) \sin(\pi \omega \cdot v) - \alpha \phi_Q(\omega)\ .$$

Now since $\omega$ is a frequency-witness for the direction $v$ we have that $|\sin(\pi \omega \cdot v)| \ge A$. Therefore, by the reverse triangle inequality we have that

$$|T_{\widehat{\mu}}(\omega)| \ge 2(1 - \alpha)A - \alpha\ .$$

Which concludes the proof of Claim B.3. $\qquad\square$

*Claim* B.4 (Close candidates have small $T_{\widehat{\mu}}$). Let $\widehat{\mu} \in \mathbb{R}^d$ be a candidate mean and set $v := \widehat{\mu} - \mu$. Then, $|T_{\widehat{\mu}}(\omega)| \le 2(1 - \alpha)\pi \|\omega\| \|v\| + \alpha$.

*Proof of Claim B.4.* Recall that

$$T_{\widehat{\mu}}(\omega) = (1 - \alpha)\, e^{2\pi i\, \omega \cdot \widehat{\mu}}\ -\ \phi(\omega)$$
$$= (1 - \alpha)\big(e^{2\pi i\, \omega \cdot \widehat{\mu}} - e^{2\pi i\, \omega \cdot \mu}\big)\ -\ \alpha\, \phi_Q(\omega)\ .$$

Now since $e^{2\pi i\, \omega \cdot \widehat{\mu}} - e^{2\pi i\, \omega \cdot \mu} = e^{2\pi i\, \omega \cdot (\mu + v/2)}\big(e^{\pi i\, \omega \cdot v} - e^{-\pi i\, \omega \cdot v}\big) = 2i\, e^{2\pi i\, \omega \cdot (\mu + v/2)} \sin(\pi\, \omega \cdot v)$, we get

$$|T_{\widehat{\mu}}(\omega)|\ \le\ 2(1 - \alpha)\, |\sin(\pi\, \omega \cdot v)|\ +\ \alpha\, |\phi_Q(\omega)|\ .$$

Because $|\phi_Q(\omega)| \le 1$ for all $\omega \in \mathbb{R}^d$, this yields

$$|T_{\widehat{\mu}}(\omega)|\ \le\ 2(1 - \alpha)\, |\sin(\pi\, \omega \cdot v)|\ +\ \alpha\ .$$

Finally, using that $|\sin t| \le |t|$ and $|\omega \cdot v| \le \|\omega\|\, \|v\|$, gives us

$$|T_{\widehat{\mu}}(\omega)|\ \le\ 2(1 - \alpha)\pi \|\omega\| \|v\|\ +\ \alpha\ ,$$

which concludes the proof of Claim B.4. $\qquad\square$

*Claim* B.5 (Covering over $\omega$ preserves frequency witnesses). Fix $v \in \mathbb{R}^d$. Let $\mathcal{C}_\omega$ be an $\eta$-cover of $\mathcal{B}_d(B_\delta)$. If there exists a $(\epsilon, A, \delta)$-frequency-witness for the direction $v$, then there exists an $\omega \in \mathcal{C}_\omega$ that is a $(\epsilon, A - \pi\eta \|v\|, \delta - \eta L)$-frequency-witness for $v$.

*Proof of Claim B.5.* Let $\omega_v$ be a $(\epsilon, A, \delta)$-frequency-witness for the direction $v$. By the definition of $\mathcal{C}_\omega$, there exists an $\omega \in \mathcal{C}_\omega$ such that $\|\omega_v - \omega\| \leq \eta$. Therefore, from the Lipschitzness assumption on $\phi_D$, it follows that $|\phi_D(\omega)| \geq \delta - \eta L$. Moreover, since $|\sin(\pi v \cdot \omega_v)| \geq A$, and sin is 1-Lipschitz, we obtain $|\sin(\pi v \cdot \omega)| \geq A - \pi\eta\|v\|$. Thus $\omega$ is a $(\epsilon, A - \pi\eta\|v\|, \delta - \eta L)$-frequency-witness for $v$, as desired. Which completes the proof of Claim B.5. $\square$

*Claim* B.6 (Concentration of $\widehat{T}$). Fix a sufficiently large universal constant $C > 0$ and $\eta, \tau \in (0,1)$. Let $\mathcal{C}_\omega, C_\mu$ be finite subsets of $\mathbb{R}^d$. If $n \geq C \log(|\mathcal{C}_\omega|/\tau)/(\eta\delta)^2$, then with probability at least $1 - \tau$ for any $\omega \in \mathcal{C}_\omega$ such that $|\phi_D(\omega)| \geq \delta$ and any $\widehat{\mu} \in C_\mu$ it holds that $|T_{\widehat{\mu}}(\omega) - \widehat{T}_{\widehat{\mu}}(\omega)| \leq \eta$.

*Proof of Claim B.6.* Let $x_1, \ldots, x_n$ be i.i.d. contaminated samples. For fixed $\omega$, write $y_j = e^{2\pi i \omega \cdot x_j}$ so both $\mathrm{Re}(y_j)$ and $\mathrm{Im}(y_j)$ lie in $[-1,1]$ and $\widehat{\phi}(\omega) = \frac{1}{n}\sum_{j=1}^n y_j$. Hoeffding on real and imaginary parts gives

$$\mathbf{Pr}\left[\,\left|\widehat{\phi}(\omega) - \phi(\omega)\right| \geq t\,\right] \leq 4\,e^{-nt^2/4}\,.$$

Similarly, this also holds for $\phi_D$, i.e., $\mathbf{Pr}\left[\,\left|\widehat{\phi}_D(\omega) - \phi_D(\omega)\right| \geq t\,\right] \leq 4\,e^{-mt^2/4}$.

Since $\widehat{T}_{\widehat{\mu}}(\omega) - T_{\widehat{\mu}}(\omega) = \psi(\omega) - \widehat{\psi}(\omega) = (\phi(\omega) - \widehat{\phi}(\omega))/\phi_D(\omega)$ and $|\phi_D(\omega)| \geq \delta$, the event $\left|T_{\widehat{\mu}}(\omega) - \widehat{T}_{\widehat{\mu}}(\omega)\right| \leq \eta$ holds whenever $\left|\widehat{\phi}(\omega) - \phi(\omega)\right| \leq \eta\delta$. A union bound over $\omega \in \mathcal{C}_\omega$ yields failure probability at most $4|\mathcal{C}_\omega|e^{-n\eta^2\delta^2/4}$, which we make $\leq \tau$ by the stated choice of $n$. Note the deviation is independent of $\widehat{\mu}$, so no extra $|\mathcal{C}_\mu|$ factor is needed. Which concludes the proof of Claim B.6. $\square$

# C. Omitted Content from Section 4

In this section, we provide the content omitted from our lower bound section.

*Claim* C.1 (Derivative bounds imply polynomial $L^2 - L^\infty$ relationship). Let $D$ be a distribution on $\mathbb{R}$ with density $p$ such that $p' \in L^1(\mathbb{R})$. Assume $\|p'\|_{L^1} = O(1)$, then for any measurable set $S \subseteq \mathbb{R}$ it holds $\|\phi_D \mathbb{1}_S\|_{L^2} \lesssim \sqrt{\|\phi_D \mathbb{1}_S\|_{L^\infty}}$.

*Proof.* First by Fact A.2 we have

$$\phi_D(\omega) = -\frac{1}{2\pi i\omega} \int_\mathbb{R} p'(x)e^{2\pi i\omega x}\,dx,$$

hence $|\phi_D(\omega)| \leq \|p'\|_{L^1}/(2\pi|\omega|)$ for all $\omega \neq 0$. Denote by $M := \sup_{\omega \in S} |\phi_D(\omega)|$. Now fix any $T > 0$ and write

$$\|\phi_D \mathbb{1}_S\|_{L^2}^2 = \int_{S \cap \{|\omega| \leq T\}} |\phi_D(\omega)|^2\,d\omega + \int_{S \cap \{|\omega| > T\}} |\phi_D(\omega)|^2\,d\omega.$$

The first term is at most $(2T)M^2$. For the second term, use the $1/|\omega|$ decay:

$$\int_{|\omega| > T} |\phi_D(\omega)|^2\,d\omega \leq \int_{|\omega| > T} \left(\frac{\|p'\|_{L^1}}{2\pi|\omega|}\right)^2 d\omega \lesssim \frac{\|p'\|_{L^1}^2}{T}.$$

Thus

$$\|\phi_D \mathbb{1}_S\|_{L^2}^2 \lesssim TM^2 + \frac{\|p'\|_{L^1}^2}{T}.$$

Choosing $T = \|p'\|_{L^1}/M$ yields $\|\phi_D \mathbb{1}_S\|_{L^2}^2 \lesssim M\|p'\|_{L^1}$, proving the claim. $\square$

**Lemma C.2** (Characteristic function to TV distance closeness). *Let $P$ and $Q$ be distributions over $\mathbb{R}$ with densities $p$ and $q$ respectively. Denote by $\Delta\phi := \phi_P - \phi_Q$ the difference of their characteristic functions. Then, for every $R > 0$,*

$$d_{\mathrm{TV}}(P, Q) \leq \frac{1}{2}\|(p - q)\mathbb{1}\{|x| > R\}\|_{L^1} + \sqrt{\frac{R}{2}}\,\|\Delta\phi\|_{L^2}\,.$$

*Proof.* Define $h := p - q$. Note that $d_{\mathrm{TV}}(P,Q) = \|h\|_{L^1}/2$. For $R > 0$, we write $\|h\|_{L^1} = I_{\mathrm{out}} + I_{\mathrm{in}}$, where $I_{\mathrm{out}} := \int_{|x|>R} |h(x)|\,dx$ and $I_{\mathrm{in}} := \int_{|x|\leq R} |h(x)|\,dx$. We leave the outside region contribution, $I_{\mathrm{out}}$, as is.

For the inside region contribution we have that by Cauchy–Schwarz, $I_{\text{in}} \leq (2R)^{1/2} \|h\|_{L^2}$. Using Plancherel's theorem this yields
$$I_{\text{in}} \leq (2R)^{1/2} \|\Delta\phi\|_{L^2}.$$

Which completes the proof of Lemma C.2. $\qquad\square$

**Lemma C.3** (Fourier Matching). *Let $\epsilon, \alpha \in (0,1)$, $\widehat{f}(\omega) := (1-\alpha)(e^{\pi i \epsilon \omega} - e^{-\pi i \epsilon \omega})/\alpha$ and $f := \mathcal{F}^{-1}[\widehat{f}]$. For every $w > 0$ there exists a function $\widehat{\rho}_w : \mathbb{R} \to \mathbb{C}$ with $\widehat{\rho}_w(\omega) = 1$ for all $\omega : |\omega - i/\epsilon| \leq w$ for some $i \in \mathbb{Z}$, $\widehat{g} := \widehat{f} \cdot \widehat{\rho}_w$, and $g := \mathcal{F}^{-1}[\widehat{g}]$, such that the following hold:*

1. *$g$ is a real valued signed measure.*

2. *$\int_{-\infty}^{\infty} g(x)\,dx = 0$.*

3. *$\|g(x)\|_{L^1} \lesssim \epsilon w/\alpha$.*

4. *For every $R > \max\{2\epsilon, 2/w\}$ it holds that $\|g\mathbb{1}\{|x| > R\}\|_{L^1} \lesssim \frac{\epsilon}{\alpha} \frac{1}{w^2 R^3}$.*

*Proof of Lemma C.3.* Let $\widehat{b}_w : \mathbb{R} \to \mathbb{C}$ be the window function obtained by convolving 4 rectangular windows (recall that $\chi_w(\omega) := \mathbb{1}\{\omega \in [-w, w]\}$), one of width $3w/2$ and three of width $w/6$, and then normalizing the resulting convoluted function:
$$\widehat{b}_w(\omega) = \left(\frac{3}{w}\right)^3 (\chi_{3w/2} * \chi_{w/6} * \chi_{w/6} * \chi_{w/6})(\omega).$$

*Claim* C.4 (Window Function Properties). The function $\widehat{b}_w$ satisfies the following properties:

1. $\widehat{b}_w$ is an even function.

2. $\widehat{b}_w(\omega) = 1$ for all $\omega \in [-w, w]$ and $\widehat{b}_w(\omega) = 0$ for all $\omega \notin [-2w, 2w]$.

*Proof of Claim C.4.* First, note that $\widehat{b}_w$ is even as each component of the convolution is even.

Next, note that for $r, s > 0$ we have that
$$(\chi_r * \chi_s)(\omega) = \int_{\omega-s}^{\omega+s} \chi_r(t)dt.$$

Therefore, $\text{supp}(\chi_r * \chi_s) = [-(r+s), r+s]$. If $r \geq s$, then for $|\omega| \leq r - s$ the limits of the above integration lie fully inside $[-r, r]$. Hence, the integral is constantly equal to $2s$ for al for $|\omega| \leq r - s$, i.e. $(\chi_r * \chi_s)(\omega) = 2s$ for $|\omega| \leq r - s$.

We apply this reasoning iteratively. Let $a = 3w/2$ and $b = c = d = w/6$. First $h_1 := \chi_a * \chi_b$ has support $|\omega| \leq a + b$ and a flat plateau of height $2b$, for $|\omega| \leq a - b$. Next $h_2 := h_1 * \chi_c$ has support $|\omega| \leq a + b + c$ and, since $h_1$ is constant on $|\omega| \leq a - b$, $h_2$ has a flat plateau of height $(2b)(2c)$, for $|\omega| \leq a - b - c$. Finally, $h_3 := h_2 * \chi_d$ has support $|\omega| \leq a + b + c + d$ and a flat plateau of height $(2b)(2c)(2d) = 8bcd$ on $|\omega| \leq a - b - c - d$.

With $a = 3w/2$ and $b = c = d = w/6$, we have $a + b + c + d = 2w$ and $a - b - c - d = w$, hence
$$\text{supp}(h_3) = [-2w, 2w] \quad \text{and} \quad h_3(x) = 8bcd \text{ on } [-w, w].$$

Since $8bcd = 8(w/6)^3 = w^3/27$, the stated normalization gives
$$\widehat{b}_w(x) = (3/w)^3 h_3(x) = 1 \text{ on } [-w, w], \qquad \widehat{b}_w(x) = 0 \text{ for } |x| > 2w.$$

This completes the proof of Claim C.4. $\qquad\square$

Write $b_w = \mathcal{F}^{-1}[\widehat{b}_w]$. Since $\widehat{b}_w$ is a convolution of rectangular windows and the inverse Fourier transform of a rectangular window $\mathbb{1}\{\omega \in [-w, w]\}$ is $2w \, \text{sinc}(2w\omega)$, we have that $b_w(x) = 3^3 2^4 w \, \text{sinc}(3wx) \, \text{sinc}^3(wx/3)$.

Define the $1/\epsilon$–periodized window
$$\widehat{\rho}_w(\omega) := \sum_{m \in \mathbb{Z}} \widehat{b}_w(\omega - m/\epsilon),$$

and set $\widehat{g}(\omega) := \widehat{f}(\omega)\,\widehat{\rho}_w(\omega)$, $g := \mathcal{F}^{-1}[\widehat{g}]$.

Note that since $\widehat{g}(\omega)$ is periodic with period $2/\epsilon$ ($\widehat{f}$ has period $2/\epsilon$ and $\widehat{\rho}_w$ has period $1/\epsilon$), we have that $g$ is a collection of $\delta$-functions (see Fact A.1).

First, we prove that $g$ is a real signed measure. Since $\widehat{\rho}_w$ is even and real valued, and $\widehat{f}(-\omega) = \overline{\widehat{f}(\omega)}$ (indeed $\widehat{f}(\omega) = 2(1-\alpha)i\sin(\pi\omega\epsilon)/\alpha$ ), we get $\widehat{g}(-\omega) = \widehat{f}(-\omega)\widehat{\rho}_w(-\omega) = \overline{\widehat{f}(\omega)}\widehat{\rho}_w(\omega) = \overline{\widehat{g}(\omega)}$. Hence, by Fact A.1 we have that $g$ is a real valued measure, i.e., the coefficients of the $\delta$'s are real-valued. This proves the first item of Lemma 4.6.

Second, note that by the definition of the Fourier transform we have that $\int_{-\infty}^{\infty} g(x)dx = \widehat{g}(0) = \widehat{f}(0)\widehat{\rho}_w(0)$. Now since $\widehat{f}(0) = 0$, we have that $\int_{-\infty}^{\infty} g(x)dx = 0$. This proves the second item of Lemma 4.6.

Now we bound the $L^1$ norm of $g$. To compute the $L^1$ norm it suffices to compute the coefficients of the $\delta$-functions. To compute the inverse $g$ note that $\widehat{\rho}_w$ is a convolution of $\text{comb}_{1/\epsilon}(\omega) := \sum_{m\in\mathbb{Z}} \delta(\omega - m/\epsilon)$ and $\widehat{b}_w$ with $\mathcal{F}^{-1}\left[\text{comb}_{1/\epsilon}\right](x) = \epsilon\sum_{n\in\mathbb{Z}} \delta(x - n\epsilon)$. As a result the inverse fourier transform of $\widehat{\rho}_w$ is

$$\rho_w(x) = \epsilon\sum_{n\in\mathbb{Z}} b_w(n\epsilon)\,\delta(x - n\epsilon).$$

Finally to compute $g$, we can just convolve $f$ and $\rho_w$ which gives us

$$g(x) = (1-\alpha)\frac{\epsilon}{\alpha}\sum_{k\in\mathbb{Z}}\left(b_w((k+1)\epsilon) - b_w(k\epsilon)\right)\delta(x - k\epsilon - \epsilon/2).$$

Therefore, we have that

$$\|g\|_{L^1} = (1-\alpha)\frac{\epsilon}{\alpha}\sum_{k\in\mathbb{Z}}\left|b_w((k+1)\epsilon) - b_w(k\epsilon)\right|.$$

By the fundamental theorem of calculus and the triangle inequality, for each $k \in \mathbb{Z}$,

$$\left|b_w((k+1)\epsilon) - b_w(k\epsilon)\right| = \left|\int_{k\epsilon}^{(k+1)\epsilon} b'_w(x)\,dx\right|$$
$$\leq \int_{k\epsilon}^{(k+1)\epsilon} |b'_w(x)|\,dx.$$

Summing over $k \in \mathbb{Z}$ gives us

$$\sum_{k\in\mathbb{Z}}\left|b_w((k+1)\epsilon) - b_w(k\epsilon)\right| \leq \int_{\mathbb{R}}|b'_w(x)|\,dx.$$

Substituting the derivative and changing variables gives us

$$\int_{\mathbb{R}}|b'_w(x)|\,dx$$
$$\lesssim 3w\int_{\mathbb{R}}\Big(|\text{sinc}'(3y)|\,|\text{sinc}(y/3)|^3$$
$$+ |\text{sinc}(3y)|\,|\text{sinc}(y/3)|^2\,|\text{sinc}'(y/3)|\Big)\,dy.$$

We use the following bounds $|\text{sinc}(t)| \leq \min\{1, 1/(\pi|t|)\}$, $\text{sinc}'(x) \leq \min\{\sqrt{2/3}\pi, 2/|t|\}$ $t \in \mathbb{R}$. Applying this bound we get that

$$\int_{\mathbb{R}}|b'_w(x)|\,dx \lesssim w.$$

Hence,

$$\|g\|_{L^1} = (1-\alpha)\frac{\epsilon}{\alpha}\sum_{k\in\mathbb{Z}}\left|b_w((k+1)\epsilon) - b_w(k\epsilon)\right| \lesssim \frac{\epsilon}{\alpha}w .$$

Now it suffices to bound the tail behavior of $g$. Fix $R > 0$. Similarly to the above we have

$$
\begin{aligned}
&\|g\mathbb{1}\{|x| > R\}\|_{L^1} \\
&= (1 - \alpha)\frac{\epsilon}{\alpha} \sum_{k: |k\epsilon + \epsilon/2| > R} \left|b_w((k+1)\epsilon) - b_w(k\epsilon)\right|.
\end{aligned}
$$

Using the fundamental theorem of calculus and summing only over those $k$ with $|k\epsilon + \epsilon/2| > R$, we obtain

$$
\|g\mathbb{1}\{|x| > R\}\|_{L^1} \leq (1 - \alpha)\frac{\epsilon}{\alpha} \int_{|x| > R - \epsilon} |b'_w(x)|\, dx.
$$

Recall that $b_w(x) = C_0\, w\, \mathrm{sinc}(3wx)\, \mathrm{sinc}^3(wx/3)$ for a universal constant $C_0$. Differentiating and changing variables $y = wx$ gives

$$
\begin{aligned}
&\int_{|x| > R - \epsilon} |b'_w(x)|\, dx \\
&\lesssim w \int_{|y| > w(R - \epsilon)} \Big( |\mathrm{sinc}'(3y)|\,|\mathrm{sinc}(y/3)|^3 \\
&\qquad\qquad + |\mathrm{sinc}(3y)|\,|\mathrm{sinc}(y/3)|^2\,|\mathrm{sinc}'(y/3)| \Big)\, dy.
\end{aligned}
$$

Now using the tail bound for $\mathrm{sinc}$ similarly to the proof of (3) we have, for $w(R - \epsilon) \geq 1$,

$$
\begin{aligned}
\int_{|x| > R - \epsilon} |b'_w(x)|\, dx &\lesssim w \int_{|y| > w(R - \epsilon)} \frac{dy}{|y|^4} \\
&\lesssim \frac{1}{w^2\,(R - \epsilon)^3} \lesssim \frac{1}{w^2 R^3}.
\end{aligned}
$$

Substituting back gives

$$
\|g\mathbb{1}\{|x| > R\}\|_{L^1} \lesssim \frac{\epsilon}{\alpha}\frac{1}{w^2 R^3}\ .
$$

Which completes the proof of Lemma C.3 $\hfill\square$

## D. Applications to Well-Known Distributions

In this section, we provide the applications of our theorems to several well-studied distribution families. We summarize our results in Table 1. We begin by instantiating the upper bound, Theorem 3.2.

**Corollary D.1** (Estimating the mean of a Standard Gaussian). *Let $\alpha \in (0, 1/4), \epsilon \in (0, 1), d \in \mathbb{Z}_+$. There exists an algorithm that given $\widetilde{O}(d\, e^{O((\alpha/\epsilon)^2)})$ i.i.d. $\alpha$–mean–shift contaminated samples from $\mathcal{N}(\mu, I_d)$ returns an estimate $\widehat{\mu}$ such that $\|\widehat{\mu} - \mu\| \leq \epsilon$ with probability at least $2/3$.*

*Proof.* First, using $d/\alpha^2$ samples and polynomial time we estimate the mean to $\ell_2$-error $O(1)$ using an adversarially robust estimator (see Fact A.4). We use this estimate to recenter the distribution so that $\|\mu\| = O(1)$. Following we verify the conditions of Theorem 3.2 for the Gaussian.

For $D = \mathcal{N}(0, I_d)$ we have $\phi_D(\omega) = e^{-2\pi^2\|\omega\|^2}$. Hence $\nabla\phi_D(\omega) = -4\pi^2\,\omega\,e^{-2\pi^2\|\omega\|^2}$ and thus $\|\nabla\phi_D(\omega)\| \leq 4\pi^2\|\omega\|e^{-2\pi^2\|\omega\|^2} = O(1)$, so $\phi_D$ is $L$–Lipschitz with $L = O(1)$.

Now fix $v \in \mathbb{R}^d$ with $\|v\| \geq \epsilon$ and set $A := 4\alpha$ and $\omega := \frac{\arcsin(A)}{\pi}\frac{v}{\|v\|}$. Then $|\sin(\pi v \cdot \omega)| = \sin(\arcsin(A)) = A$ and $\|\omega\| = \frac{\arcsin(A)}{\pi\|v\|} \lesssim \alpha/\epsilon$. Therefore $|\phi_D(\omega)| = e^{-2\pi^2\|\omega\|^2} \geq e^{-c(\alpha/\epsilon)^2}$ for a universal constant $c > 0$. Setting $\delta := e^{-c(\alpha/\epsilon)^2}$, we conclude that $D$ satisfies the $(\epsilon, A, \delta)$-frequency-witness condition.

The density of $D$ is $p_D(x) = (2\pi)^{-d/2}e^{-\|x\|^2/2}$. For each $j \in [d]$ we have $\frac{\partial}{\partial x_j}p_D(x) = -x_j p_D(x)$, so $\frac{\partial}{\partial x_j}p_D \in L^1(\mathbb{R}^d)$ and

$$
\left\|\frac{\partial}{\partial x_j}p_D\right\|_{L^1(\mathbb{R}^d)} = \int_{\mathbb{R}^d} |x_j|p_D(x)\, dx = \mathop{\mathbf{E}}_{x \sim \mathcal{N}(0, I_d)}[|x_j|] = \sqrt{\frac{2}{\pi}}\ .
$$

Hence, $M_1 := \max_{j \in [d]} \left\| \frac{\partial}{\partial x_j} p_D \right\|_{L^1(\mathbb{R}^d)} = \sqrt{\frac{2}{\pi}} = O(1)$.

Finally, applying Theorem 3.2 with $L = O(1)$, $A = 4\alpha$, and $\delta = e^{-c(\alpha/\epsilon)^2}$ yields sample complexity

$$n = O\left( d \log\left( \frac{\sqrt{d}\, M_1 R L}{\delta^2 A} \right) \frac{1}{(((1-\alpha)A - 2\alpha)\delta)^2} \right)$$
$$= \widetilde{O}(d\, e^{O((\alpha/\epsilon)^2)}),$$

and Algorithm 1 returns $\widehat{\mu}$ with $\|\widehat{\mu} - \mu\| \leq \epsilon$ with probability at least $2/3$. $\qquad\square$

**Corollary D.2** (Estimating the mean of a Standard Laplace)**.** *Let $\alpha \in (0, 1/4)$, $\epsilon \in (0, 1)$, and $d \in \mathbb{Z}_+$. There exists an algorithm that, given $n = \widetilde{O}\left( d\, \frac{\alpha^2}{\epsilon^4} \right)$ i.i.d. $\alpha$–mean–shift contaminated samples from the $d$–dimensional product Laplace distribution with unit covariance (i.e., each coordinate $\mathrm{Lap}(0, 1/\sqrt{2})$), returns an estimate $\widehat{\mu}$ such that $\|\widehat{\mu} - \mu\| \leq \epsilon$ with probability at least $2/3$.*

*Proof.* As in the Gaussian corollary, since $\alpha < 1/4$ using $O(d)$ samples and polynomial time, we obtain an $O(1)$-accurate robust estimate of $\mu$ (see Fact A.4) and recenter so that $\|\mu\| = O(1)$. For $D = \mathrm{Lap}_d(0, I_d)$ (i.i.d. coordinates with $b = 1/\sqrt{2}$), the characteristic function is $\phi_D(\omega) = \prod_{j=1}^{d}(1 + 2\pi^2\omega_j^2)^{-1}$.

Fix any $v$ with $\|v\| \geq \epsilon$ and set $A := 4\alpha$. Choose $\omega = \frac{\arcsin(A)}{\pi} \frac{v}{\|v\|}$, so $|\sin(\pi v \cdot \omega)| = \sin(\arcsin A) = A$ and $\|\omega\| \leq (\arcsin A)/(\pi\epsilon) = \Theta(\alpha/\epsilon)$.

Let $\delta := \frac{1}{1 + 2\pi^2\|\omega\|^2}$. Since $\|\omega\| = \Theta(\alpha/\epsilon)$ we have $\delta = \Theta\left(\frac{1}{1+(\alpha/\epsilon)^2}\right)$ and moreover $|\phi_D(\omega)| \geq \prod_{j=1}^{d} \frac{1}{1+2\pi^2\omega_j^2} \geq \frac{1}{1+2\pi^2\|\omega\|^2} = \delta$. Also $\phi_D$ is $L$-Lipschitz on $\|\xi\| \leq \|\omega\|$ with $L \lesssim 2\pi^2\|\omega\| = \widetilde{O}(\alpha/\epsilon)$. Therefore $D$ satisfies the $(\epsilon, A, \delta)$-frequency-witness condition.

The one-dimensional Laplace density with $b = 1/\sqrt{2}$ is $p_1(x) = \frac{1}{\sqrt{2}} e^{-\sqrt{2}|x|}$ and the $d$-dimensional product density is $p_D(x) = \prod_{j=1}^{d} p_1(x_j)$. For $j \in [d]$ and all $x$ with $x_j \neq 0$,

$$\frac{\partial}{\partial x_j} p_D(x) = \left( \frac{p_1'(x_j)}{p_1(x_j)} \right) p_D(x) = -\sqrt{2}\, \mathrm{sign}(x_j)\, p_D(x),$$

so $\frac{\partial}{\partial x_j} p_D \in L^1(\mathbb{R}^d)$ and

$$\left\| \frac{\partial}{\partial x_j} p_D \right\|_{L^1(\mathbb{R}^d)} = \int_{\mathbb{R}^d} \sqrt{2}\, p_D(x)\, dx = \sqrt{2} \,.$$

Hence $M_1 := \max_{j \in [d]} \left\| \frac{\partial}{\partial x_j} p_D \right\|_{L^1(\mathbb{R}^d)} = \sqrt{2} = O(1)$. Finally, applying Theorem 3.2 with $A = 4\alpha$ and $\delta = \Theta\left(\frac{1}{1+(\alpha/\epsilon)^2}\right)$ yields

$$n = O\left( d \log\left( \frac{\sqrt{d}\, M_1 R L}{\delta^2 A} \right) \frac{1}{(((1-\alpha)A - 2\alpha)\delta)^2} \right)$$
$$= \widetilde{O}\left( d\, \frac{\alpha^2}{\epsilon^4} \right),$$

and running Algorithm 1 returns $\widehat{\mu}$ with $\|\widehat{\mu} - \mu\| \leq \epsilon$ with probability at least $2/3$. $\qquad\square$

**Corollary D.3** (Estimating the mean of a Uniform distribution)**.** *Let $\alpha \in (0, 1/4)$ and $\epsilon \in (0, 1)$. Let $D$ be the uniform distribution on $[-1, 1]$ (with density $p_D(x) = \frac{1}{2}\mathbb{1}\{|x| \leq 1\}$), and let $D_\mu$ denote its shift by an unknown mean $\mu \in \mathbb{R}$. There exists an algorithm that, given $n = \widetilde{O}\left(\frac{1}{\epsilon^2}\right)$ i.i.d. $\alpha$–mean–shift contaminated samples from $D_\mu$, returns an estimate $\widehat{\mu}$ such that $|\widehat{\mu} - \mu| \leq \epsilon$ with probability at least $2/3$.*

*Proof.* As in the Gaussian/Laplace corollaries, using $O(1 =)$ samples and polynomial time, we can obtain an $O(1)$-accurate robust estimate of $\mu$ (Fact A.4) and recenter so that $|\mu| = O(1)$.

For $D = \mathrm{Unif}([-1, 1])$ we have for every $\omega \in \mathbb{R}$, the characteristic function is

$$\phi_D(\omega) = \mathop{\mathbf{E}}_{x \sim D}\big[e^{2\pi i \omega x}\big] = \frac{1}{2} \int_{-1}^{1} e^{2\pi i \omega x}\, dx$$
$$= \frac{\sin(2\pi\omega)}{2\pi\omega} = \mathrm{sinc}(2\omega).$$

Differentiation gives

$$\phi_D'(\omega) = \frac{2\pi\omega \cos(2\pi\omega) - \sin(2\pi\omega)}{2\pi\omega^2},$$

hence $\sup_{\omega \in \mathbb{R}} |\phi_D'(\omega)| = O(1)$, thus $\phi_D$ is $L$–Lipschitz with $L = O(1)$.

Fix any $v \in \mathbb{R}$ with $|v| \geq \epsilon$. Consider the interval $I_v := \left[\frac{1}{4|v|}, \frac{3}{4|v|}\right]$. For any $\omega \in I_v$ we have $|v\omega| \in [1/4, 3/4]$, and therefore $|\sin(\pi v \omega)| \geq \sin(\pi/4) = \frac{1}{\sqrt{2}}$. Set $A := 1/\sqrt{2}$ (note that $(1 - \alpha)A - 2\alpha > 0$ for all $\alpha < 1/4$).

We now lower bound $|\phi_D(\omega)|$ for a suitable choice of $\omega \in I_v$. If $|v| \geq 1$, then $\omega_0 := 1/(4|v|) \in [0, 1/4]$, and using $\sin t \geq (2/\pi)t$ for $t \in [0, \pi/2]$,

$$|\phi_D(\omega_0)| = \frac{\sin(2\pi\omega_0)}{2\pi\omega_0} \geq \frac{(2/\pi) \cdot 2\pi\omega_0}{2\pi\omega_0} = \frac{2}{\pi}.$$

If instead $|v| \in [\epsilon, 1)$, then $|I_v| = 1/(2|v|) \geq 1/2$; since $\omega \mapsto \sin(2\pi\omega)$ has period $1$, there exists $\omega_1 \in I_v$ with $|\sin(2\pi\omega_1)| \geq 1/2$. For this $\omega_1$ we have $|\omega_1| \leq 3/(4|v|)$ and hence

$$|\phi_D(\omega_1)| = \frac{|\sin(2\pi\omega_1)|}{2\pi|\omega_1|} \geq \frac{1/2}{2\pi \cdot (3/(4|v|))} = \frac{|v|}{3\pi} \geq \frac{\epsilon}{3\pi}.$$

Combining the two cases, for every $v$ with $|v| \geq \epsilon$ there exists $\omega \in I_v$ such that

$$|\sin(\pi v \omega)| \geq A \qquad \text{and} \qquad |\phi_D(\omega)| \geq \delta$$

with $\delta := \epsilon/(3\pi)$. In particular, $D$ satisfies the $(\epsilon, A, \delta)$-frequency-witness condition (see Definition 3.1) with $A = 1/\sqrt{2}$ and $\delta = \epsilon/(3\pi)$.

Also $p_D$ has a distributional derivative $\frac{d}{dx}p_D = \frac{1}{2}\delta(x + 1) - \frac{1}{2}\delta(x - 1)$. Its total variation is

$$M_1 := \left\|\frac{d}{dx}p_D\right\|_{L^1} = \tfrac{1}{2} + \tfrac{1}{2} = 1.$$

Applying Theorem 3.2 we have that with

$$n = \widetilde{O}\left(\frac{1}{((1 - \alpha)A - 2\alpha)^2\, \delta^2}\right) = \widetilde{O}\left(\frac{1}{\epsilon^2}\right),$$

samples Algorithm 1 returns $\widehat{\mu}$ with $|\widehat{\mu} - \mu| \leq \epsilon$ with probability at least $2/3$.

$\square$

**Corollary D.4** (Estimating the mean of a sum of $m$ Uniforms). *Let $\alpha \in (0, 1/8)$, $\epsilon \in (0, \alpha)$, and $m \in \mathbb{Z}_+$. Let $U_1, \ldots, U_m$ be i.i.d. $\mathrm{Unif}([-1, 1])$, let $D^{(m)}$ be the distribution of $\sum_{i=1}^{m} U_i$, and let $D_\mu^{(m)}$ denote its shift by an unknown mean $\mu \in \mathbb{R}$. There exists an algorithm that, given $n = \widetilde{O}\left(\alpha^{-2}(O(\alpha/\epsilon))^{2m}\right)$, i.i.d. $\alpha$–mean–shift contaminated samples from $D_\mu^{(m)}$, returns an estimate $\widehat{\mu}$ such that $|\widehat{\mu} - \mu| \leq \epsilon$ with probability at least $2/3$.*

*Proof.* As in the previous corollaries, since $\alpha < 1/8$ using $O(1)$ samples and polynomial time we obtain a robust estimate of $\mu$ (Fact A.4) to absolute error $O(\sqrt{m})$ (since $\mathrm{Var}(\sum_{i=1}^{m} U_i) = m/3$), and recenter so that $|\mu| = O(\sqrt{m})$.

For $D^{(m)}$ we have, for every $\omega \in \mathbb{R}$,

$$\phi_{D^{(m)}}(\omega) = \prod_{i=1}^{m} \mathbf{E}\left[e^{2\pi i \omega U_i}\right] = \left(\frac{\sin(2\pi\omega)}{2\pi\omega}\right)^m = \mathrm{sinc}(2\omega)^m.$$

Write $s(\omega) := \mathrm{sinc}(2\omega)$. Since $\sup_{\omega \in \mathbb{R}} |s'(\omega)| = O(1)$ and $|s(\omega)| \le 1$,

$$\sup_{\omega \in \mathbb{R}} \left|\phi'_{D^{(m)}}(\omega)\right| = \sup_{\omega \in \mathbb{R}} \left|m\, s(\omega)^{m-1} s'(\omega)\right| = O(m),$$

so $\phi_{D^{(m)}}$ is $L$–Lipschitz with $L = O(m)$.

Fix any $v \in \mathbb{R}$ with $|v| \ge \epsilon$ and set $A := 4\alpha$ (so $(1-\alpha)A - 2\alpha = 2\alpha(1-2\alpha) > 0$). Let

$$\omega_v^\star := \frac{\arcsin(A)}{\pi v}, \qquad \text{so that} \qquad \left|\sin(\pi v \omega_v^\star)\right| = A.$$

We now show that for every $|v| \ge \epsilon$ there exists $\omega$ with

$$|\sin(\pi v \omega)| \ge A \qquad \text{and} \qquad |\phi_{D^{(m)}}(\omega)| \ge \left(O\left(\frac{|v|}{A}\right)\right)^m.$$

This implies the $(\epsilon, A, \delta)$-frequency-witness condition with

$$\delta \ge \left(O\left(\frac{\epsilon}{\alpha}\right)\right)^m.$$

If $|v| \ge 4A$, then

$$|\omega_v^\star| = \frac{\arcsin(A)}{\pi|v|} \le \frac{A}{|v|} \le \frac{1}{4},$$

and using $|\sin t| \ge (2/\pi)|t|$ for $|t| \le \pi/2$ we get

$$\left|\frac{\sin(2\pi\omega_v^\star)}{2\pi\omega_v^\star}\right| \ge \frac{2}{\pi}, \qquad \text{hence} \qquad |\phi_{D^{(m)}}(\omega_v^\star)| \ge (2/\pi)^m.$$

If $|v| < 4A$. Let $\theta := \arcsin(A) \in (0, \pi/2)$ and define

$$\ell_v := \min\left\{1, \frac{\frac{\pi}{2} - \theta}{\pi|v|}\right\}.$$

Consider the interval $I_v := [\omega_v^\star, \omega_v^\star + \ell_v]$. For any $\omega = \omega_v^\star + t$ with $t \in [0, \ell_v]$ we have

$$\pi|v|t \le \frac{\pi}{2} - \theta \implies |\sin(\pi v \omega)| = \sin(\theta + \pi|v|t) \ge \sin(\theta) = A,$$

so the *entire* interval $I_v$ satisfies $|\sin(\pi v \omega)| \ge A$.

Now choose $\omega \in I_v$ maximizing $|\sin(2\pi\omega)|$ over $I_v$. A simple periodicity/compactness argument gives the quantitative bound

$$\max_{\omega \in I_v} |\sin(2\pi\omega)| \ge \begin{cases} 1, & \ell_v \ge \frac{1}{2}, \\ \sin(\pi\ell_v) \ge 2\ell_v, & \ell_v < \frac{1}{2}, \end{cases}$$

and in either case $|\sin(2\pi\omega)| \ge \min\{1, 2\ell_v\}$. Furthermore, $A = 4\alpha < 1/2$ implies $\theta = \arcsin(A) \le \arcsin(1/2) = \pi/6$, thus $\pi/2 - \theta \ge \pi/3$. Hence, if $|v| \le 4A < 1/2$ then $(\pi/2 - \theta)/(\pi|v|) \ge (\pi/3)/(\pi \cdot 1/2) = 2/3$, so $\ell_v = \min\{1, (\pi/2 - \theta)/(\pi|v|)\} \ge 2/3$. Also $|\omega| \le |\omega_v^\star| + \ell_v \le |\omega_v^\star| + 1 = O(\omega_v^\star)$, using $|v| < 4A$ so $|\omega_v^\star| \gtrsim 1$ and hence $|\omega_v^\star| + 1 = O(|\omega_v^\star|)$.

Therefore

$$|s(\omega)| = \left|\frac{\sin(2\pi\omega)}{2\pi\omega}\right| \ge \frac{\min\{1, 2\ell_v\}}{2\pi(|\omega_v^\star| + 1)} \gtrsim \frac{|v|}{A}.$$

So

$$|\phi_{D^{(m)}}(\omega)| = |s(\omega)|^m \geq \left(\frac{c|v|}{A}\right)^m.$$

For a sufficiently small constant $c > 0$.

Combining the two cases with the fact that $\epsilon/\alpha \leq 1$ we have that for every $|v| \geq \epsilon$ there is some $\omega$ with $|\sin(\pi v\omega)| \geq A$ and

$$|\phi_{D^{(m)}}(\omega)| \geq \left(\frac{c\epsilon}{\alpha}\right)^m.$$

Thus $D^{(m)}$ satisfies the $(\epsilon, A, \delta)$-frequency-witness condition with $\delta \geq (\Omega(\epsilon/\alpha))^m$.

Now let $p_U$ be the density of the uniform distribution on $[-1, 1]$. Since $(f * g)' = f' * g = f * g'$, we have $(p_U^{*m})' = (p_U^{*(m-1)} * p_U)' = p_U^{*(m-1)} * p_U'$. Then $\|(p_U^{*m})'\|_1 = \|p_U^{*(m-1)} * p_U'\|_1 \leq \|p_U^{*(m-1)}\|_1 \|p_U'\|_1 = 1 \cdot \|p_U'\|_1 \leq 1$. This gives $M_1 \leq 1$ for the density of $D^{(m)}$. Applying Theorem 3.2 yields that with

$$n = \widetilde{O}\left(\frac{1}{(((1-\alpha)A - 2\alpha)^2)\,\delta^2}\right) = \widetilde{O}\left(\frac{C^m}{\alpha^2} \cdot \frac{\alpha^{2m}}{\epsilon^{2m}}\right) = \widetilde{O}\left(\frac{C^m \alpha^{2m-2}}{\epsilon^{2m}}\right),$$

for a sufficiently large constant $C > 0$ universal constant, Algorithm 1 returns $\widehat{\mu}$ with $|\widehat{\mu} - \mu| \leq \epsilon$ with probability at least $2/3$. $\qquad\square$

Moreover, using Theorem 4.1 yields the following up to polynomial factors tight lower bounds.

**Corollary D.5** (Standard Gaussian lower bound). *Fix $\alpha, \epsilon \in (0, 1/2)$, $\epsilon < \alpha$ with $\epsilon/\alpha < c$ for a sufficiently small universal constant $c$. Then any algorithm that estimates the mean of a standard Gaussian to absolute error $\epsilon$ with probability at least $2/3$ from $\alpha$-mean-shift contaminated samples (as in Definition 1.1), must use at least $\Omega(e^{\Omega((\alpha/\epsilon)^2)})$ , samples.*

*Proof.* The characteristic function of $D = \mathcal{N}(0, 1)$ is

$$\phi_D(\omega) = \exp(-2\pi^2\omega^2).$$

Let $S := \{\omega : \text{dist}(\epsilon\omega, \mathbb{Z}) > c\alpha\}$ (for the constant $c > 0$ from Theorem 4.1). Note that $S \subseteq \{|\omega| > c\alpha/\epsilon\}$, since $|\omega| \leq c\alpha/\epsilon$ implies $\text{dist}(\epsilon\omega, \mathbb{Z}) \leq |\epsilon\omega| \leq c\alpha$. Hence

$$\left\|\phi_D(\omega)\mathbb{1}\{\omega \in S\}\right\|_{L^2}^2 \leq \int_{|\omega|>c\alpha/\epsilon} e^{-4\pi^2\omega^2}\, d\omega$$

$$= 2\int_{c\alpha/\epsilon}^{\infty} e^{-4\pi^2\omega^2}\, d\omega \lesssim \frac{\epsilon}{\alpha} \exp\left(-4\pi^2 c^2\left(\frac{\alpha}{\epsilon}\right)^2\right).$$

Therefore,

$$\delta := \left\|\phi_D(\omega)\mathbb{1}\{\omega \in S\}\right\|_{L^2} \lesssim \sqrt{\frac{\epsilon}{\alpha}} \exp\left(-2\pi^2 c^2\left(\frac{\alpha}{\epsilon}\right)^2\right).$$

For Condition (2) of Theorem 4.1, let $\sigma := \mathbf{E}_{x\sim D}[|x|] = \sqrt{2/\pi}$. By Markov's inequality, for all $R > 0$,

$$\mathbf{Pr}_{x\sim D}[|x| \geq R] \leq \sigma/R.$$

Also, since $\delta$ is exponentially small in $(\alpha/\epsilon)^2$ and $\sigma = \Theta(1)$, the requirement $\sigma > C\delta$ in Theorem 4.1 holds (for the universal constant $C$).

Applying Theorem 4.1 gives

$$n = \Omega\left(\frac{1}{(\delta\sigma)^{1/3} + (\epsilon/\alpha)^3(\delta/\sigma)^2}\right) = \Omega\left(\frac{1}{\delta^{1/3}}\right) = \Omega\left(e^{\Omega((\alpha/\epsilon)^2)}\right),$$

where the last step uses the above bound on $\delta$ (the prefactor $\sqrt{\epsilon/\alpha}$ is absorbed in the exponent). $\qquad\square$

**Corollary D.6** (Laplace lower bound). *Fix $\alpha, \epsilon \in (0, 1/2)$, $\epsilon < \alpha$ with $\epsilon/\alpha < c$ for a sufficiently small universal constant $c$. Let $D = \mathrm{Lap}(0, 1)$ be the standard Laplace distribution with density $p(x) = \frac{1}{2}e^{-|x|}$. Then any algorithm that estimates the mean of a standard Laplace to absolute error $\epsilon$ with probability at least $2/3$ from $\alpha$-mean-shift contaminated samples (as in Definition 1.1), must use at least $\Omega\left(\sqrt{\alpha/\epsilon}\right)$ samples.*

*Proof.* The characteristic function of $D = \mathrm{Lap}(0, 1)$ is

$$\phi_D(\omega) = \int_{\mathbb{R}} \frac{1}{2}e^{-|x|}e^{2\pi i\omega x}\,dx = \frac{1}{1 + (2\pi\omega)^2} \ .$$

Let $S := \{\omega : \mathrm{dist}(\epsilon\omega, \mathbb{Z}) > c\alpha\}$ (for the constant $c > 0$ from Theorem 4.1). As in the Gaussian proof, $S \subseteq \{|\omega| > c\alpha/\epsilon\}$ and therefore

$$\left\|\phi_D(\omega)\mathbb{1}\{\omega \in S\}\right\|_{L^2}^2 \leq \int_{|\omega| > c\alpha/\epsilon} \frac{1}{\left(1 + (2\pi\omega)^2\right)^2}\,d\omega$$

$$= 2\int_{c\alpha/\epsilon}^{\infty} \frac{1}{\left(1 + (2\pi\omega)^2\right)^2}\,d\omega \ \lesssim\ \int_{c\alpha/\epsilon}^{\infty} \frac{d\omega}{\omega^4} \ \lesssim\ \left(\frac{\epsilon}{\alpha}\right)^3.$$

Hence,

$$\delta \ := \ \left\|\phi_D(\omega)\mathbb{1}\{\omega \in S\}\right\|_{L^2} \ \lesssim\ \left(\frac{\epsilon}{\alpha}\right)^{3/2}.$$

For Condition (2) of Theorem 4.1, let $\sigma := \mathbf{E}_{x\sim D}[|x|] = 1$. By Markov's inequality, for all $R > 0$,

$$\mathbf{Pr}_{x\sim D}[|x| \geq R] \leq \sigma/R.$$

Also, since $\epsilon < \alpha$ we have $\delta \lesssim (\epsilon/\alpha)^{3/2} < 1/c$, so the requirement $\sigma > C\delta$ in Theorem 4.1 holds (for the universal constant $C$).

Applying Theorem 4.1 gives

$$n = \Omega\left(\frac{1}{(\delta\sigma)^{1/3} \ + \ (\epsilon/\alpha)^3(\delta/\sigma)^2}\right).$$

With $\sigma = 1$ and $\delta \lesssim (\epsilon/\alpha)^{3/2}$, we have $(\delta\sigma)^{1/3} \lesssim (\epsilon/\alpha)^{1/2}$ and $(\epsilon/\alpha)^3(\delta/\sigma)^2 \lesssim (\epsilon/\alpha)^6$, hence the first term dominates, yielding

$$n = \Omega\left(\delta^{-1/3}\right) = \Omega\left(\left(\frac{\alpha}{\epsilon}\right)^{1/2}\right),$$

as claimed. $\qquad\square$

**Corollary D.7** (Uniform lower bound). *Fix $\alpha, \epsilon \in (0, 1/2)$, $\epsilon < \alpha$ with $\epsilon/\alpha < c$ for a sufficiently small universal constant $c$. Let $D = \mathrm{Unif}([-1, 1])$ and let $D_\mu$ denote its shift by an unknown mean $\mu \in \mathbb{R}$. Then any algorithm that estimates the mean to absolute error $\epsilon$ with probability at least $2/3$ from $\alpha$-mean-shift contaminated samples (as in Definition 1.1) must use at least $\Omega\left((\alpha/\epsilon)^{1/6}\right)$ samples.*

*Proof.* For $D = \mathrm{Unif}([-1, 1])$ we have

$$\phi_D(\omega) = \mathbf{E}_{x\sim D}\left[e^{2\pi i\omega x}\right] = \frac{\sin(2\pi\omega)}{2\pi\omega}.$$

Let $S := \{\omega : \mathrm{dist}(\epsilon\omega, \mathbb{Z}) > c\alpha\}$ (for the constant $c > 0$ from Theorem 4.1). As before, $S \subseteq \{|\omega| > c\alpha/\epsilon\}$. Hence,

$$\left\|\phi_D(\omega)\mathbb{1}\{\omega \in S\}\right\|_{L^2}^2 \leq 2\int_{c\alpha/\epsilon}^{\infty} \left(\frac{\sin(2\pi\omega)}{2\pi\omega}\right)^2 d\omega$$

$$\leq 2\int_{c\alpha/\epsilon}^{\infty} \frac{d\omega}{(2\pi\omega)^2} \lesssim \frac{\epsilon}{\alpha}.$$

Therefore, taking

$$\delta \ := \ \big\|\phi_D(\omega)\mathbb{1}\{\omega \in S\}\big\|_{L^2} \ \lesssim \ \Big(\frac{\epsilon}{\alpha}\Big)^{1/2}$$

verifies Condition (1) of Theorem 4.1.

For Condition (2), since $|x| \leq 1$ almost surely under $x \sim D$, we have for all $R > 0$,

$$\mathbf{Pr}_{x \sim D}[|x| \geq R] \leq \frac{1}{R},$$

so the tail condition holds with $\sigma := 1$ (and in particular $\sigma > C\delta$ in the regime where $\epsilon/\alpha < c$ for a sufficiently small constant $c > 0$).

Applying Theorem 4.1 (with $\sigma = 1$) yields

$$n = \Omega\Big(\frac{1}{\delta^{1/3} + (\epsilon/\alpha)^3\delta^2}\Big) = \Omega\big(\delta^{-1/3}\big) = \Omega\Big(\Big(\frac{\alpha}{\epsilon}\Big)^{1/6}\Big),$$

where we used $\epsilon < \alpha$ to note $(\epsilon/\alpha)^3\delta^2 \lesssim (\epsilon/\alpha)^4 \leq \delta^{1/3}$. $\qquad\square$

**Corollary D.8** (Sum of $m$ Uniforms lower bound). *Fix $\alpha, \epsilon \in (0, 1/2)$, $\epsilon < \alpha$ with $\epsilon/\alpha < c$ for a sufficiently small universal constant $c$. Let $U_1, \ldots, U_m$ be i.i.d. $\mathrm{Unif}([-1, 1])$, let $D^{(m)}$ be the distribution of $\sum_{i=1}^m U_i$, and let $D_\mu^{(m)}$ denote its shift by an unknown mean $\mu \in \mathbb{R}$. Then any algorithm that estimates the mean to absolute error $\epsilon$ with probability at least $2/3$ from $\alpha$-mean-shift contaminated samples (as in Definition 1.1) must use at least $\Omega\big((\alpha/\epsilon)^{(2m-1)/6}\big)$ samples.*

*Proof.* For $D^{(m)}$ we have

$$\phi_{D^{(m)}}(\omega) = \Big(\frac{\sin(2\pi\omega)}{2\pi\omega}\Big)^m.$$

Let $S := \{\omega : \mathrm{dist}(\epsilon\omega, \mathbb{Z}) > c\alpha\}$ (for the constant $c > 0$ from Theorem 4.1). We have $S \subseteq \{|\omega| > c\alpha/\epsilon\}$. Using $|\sin(2\pi\omega)| \leq 1$, for $|\omega| > 0$ we have

$$|\phi_{D^{(m)}}(\omega)| \leq \Big(\frac{1}{2\pi|\omega|}\Big)^m,$$

and therefore letting $T := c\alpha/\epsilon$,

$$\big\|\phi_{D^{(m)}}(\omega)\mathbb{1}\{\omega \in S\}\big\|_{L^2}^2 \leq 2\int_T^\infty \Big(\frac{1}{2\pi\omega}\Big)^{2m} d\omega = 2\Big(\frac{1}{2\pi}\Big)^{2m}\frac{1}{(2m-1)T^{2m-1}} \lesssim \Big(\frac{\epsilon}{\alpha}\Big)^{2m-1}$$

Hence,

$$\delta \ := \ \big\|\phi_{D^{(m)}}(\omega)\mathbb{1}\{\omega \in S\}\big\|_{L^2} \ \lesssim \ \Big(\frac{\epsilon}{\alpha}\Big)^{(2m-1)/2}.$$

For Condition (2) of Theorem 4.1, we have $\mathrm{Var}_{D^{(m)}}(x) = m/3$, so $\sigma := \mathbf{E}_{D^{(m)}}[|x|] \leq \sqrt{\mathbf{E}_{D^{(m)}}[x^2]} = \sqrt{m/3}$, and Markov gives $\mathbf{Pr}_{D^{(m)}}[|x| \geq R] \leq \sigma/R$ for all $R > 0$. Since $\delta$ is exponentially small in $m$ ($\epsilon/\alpha < 1$) and $\sigma = \Theta(\sqrt{m})$, the requirement $\sigma > C\delta$ holds when $\epsilon/\alpha$ is less than a sufficiently small constant.

Applying Theorem 4.1 yields

$$n = \Omega\Big(\frac{1}{(\delta\sigma)^{1/3} + (\epsilon/\alpha)^3(\delta/\sigma)^2}\Big) = \Omega\big(\delta^{-1/3}\big) = \Omega\Big((\alpha/\epsilon)^{(2m-1)/6}\Big),$$

where we used $\epsilon < \alpha$ to note $(\epsilon/\alpha)^3(\delta/\sigma)^2 \ll \delta^{1/3}$ for all $m \geq 1$. $\qquad\square$

