# OpenReview forum: "Sample Complexity Bounds for Robust Mean Estimation with Mean-Shift Contamination"
_ICML.cc/2026/Conference — ICML 2026 regular_

### Official Review · Reviewer_4Etp · 2026-03-11

**Soundness:** 4
**Presentation:** 3
**Significance:** 3
**Originality:** 3
**Overall Recommendation:** 5
**Confidence:** 4

**Summary:**

The authors study the task of estimating the mean of a distribution in the presence of mean-shift contamination. A classical model for this problem is the Huber model: for some inlier distribution $D$ over $\mathbb{R}^d$ and a contamination proportion $\alpha \in (0,1)$, each sample $x$ is drawn from $D$ with probability $1-\alpha$ and drawn from another arbitrary outlier distribution $Q$ with probability $\alpha$. Then, the task of the estimation algorithm is to estimate the true mean of the distribution $D$ to high accuracy using a small (polynomial in $d$) number of samples.  This is a very fundamental problem in robust statistics, and has found many applications in settings where samples are only approximately i.i.d..

Unfortunately, the Huber model suffers from the drawback that for even the most basic inlier distribution (i.e. univariate Gaussian), the error must be $\Omega(\alpha)$, i.e. it is independent of the number of samples. In attempt to circumvent this and design *consistent* estimators whose error decreases as the sample size increases, there have been many works that study the sample complexity of the *mean-shift contamination model*: for $D$, $Q$ distributions over $\mathbb{R}^d$ where $D$ has mean $0$, an $\alpha$-mean-shift contaminated sample from $D_{\mu}$ is chosen by drawing a sample $x = \mu + y$ for $y \sim D$ with probability $1-\alpha$; alternatively, with probability $\alpha$ an adversary draws a shift vector $z \sim Q$ and outputs an outlier sample $x = z + y$ for $y \sim D$. While there has been recent progress on developing polynomial-time algorithms for the cases when $D$ is a Gaussian or Laplace distribution, it remained an interesting open problem to determine the conditions on the distribution $D$ under which it’s possible to do consistent mean estimation in the mean-shift contamination model efficiently. The authors answer this open problem by providing the first known algorithm which provides a consistent estimate of $\mu$ using $O(d/\delta^2)$ samples. They also show that any algorithm requires at least $1/\delta^{\Omega(1)}$ samples to recover an accurate estimate of the mean for $d= 1$ (here $\delta$ is a parameter which is roughly the largest Fourier magnitude $|\phi_D(\omega)|$ over the frequencies $\omega$ such that $\omega \cdot v$ is at least $\alpha$ away from each integer, for all mean-shift directions $||v||\geq \varepsilon$).

The proofs of the upper and lower bounds rely on Fourier analysis to decompose the observed distribution as a convolution $D * Q$, where $D$ is the $0$-centered inlier distribution and $Q$ is the mean-vector distribution. For the upper bound, they can then write the characteristic function of $D_{\mu}$ as a product of characteristic functions of $D$ and $Q$, so $\phi_{Q}(\omega)$ can be estimated as the ratio of the two other characteristic functions ( $\phi_{D_{\mu}}$ and $\phi_{D}$ can be estimated by sampling). They then show how to reduce the estimation problem to the problem of distinguishing whether a given candidate mean vector $\hat \mu$ is close or far from $\mu$, and use this as a subroutine to search for a good approximation. The lower bound exhibits two univariate hard distributions which cannot be distinguished unless one is given $1/\delta^{\Omega(1)}$ samples, and can be generalized to product distributions in $d$ dimensions.

**Compliance With Llm Reviewing Policy:**

Affirmed.

**Final Justification:**

I believe this is a significant result for the very fundamental problem of mean estimation in the presence of mean shift contamination. The authors' rebuttal addressed all of my questions. I recommend accepting this paper to ICML.

**Key Questions For Authors:**

- Is Table 1 a summary of known upper and lower bounds for Gaussian/Laplace/Uniform distributions from previous works? Perhaps it would be informative for the reader if you could add a short summary of the types of techniques which were used for those special cases, and how your approach is similar or departs from those.

**Limitations:**

Yes.

**Strengths And Weaknesses:**

Strengths:
- This is a very fundamental problem in robust statistics, and the authors provide a general answer to the open problem of characterizing the conditions under which it is possible to estimate the mean under the mean-shift contamination model. All claims are well-substantiated and technically sound.

Weaknesses:
- The upper and lower bounds are not matching in terms of $d$, $\delta$, and $\alpha$, but this is not necessarily a major weakness as this is the first paper that gives a general algorithm and lower bound for any generic inlier distribution $D$.
- The lower bound is only for product distributions in $d$ dimensions.

---

> ### Author Rebuttal · Authors · 2026-03-31
>
> We thank the reviewer for the careful reading and for the positive assessment of our paper. We are very glad that the reviewer found the problem fundamental, the results technically sound, and the overall contribution meaningful.
>
> **Upper and Lower Bound Gap**: While our upper and lower bounds do not precisely match, Remarks 4.2 and 4.3 show that they are polynomially related in their dependence in $d/\delta$. As Table 1 illustrates, this implies at most a polynomial gap for many classical distributions. Intuitively, this is because in the regime where $\alpha$ is not too close to its information-theoretic threshold (determined by $A$ in the Fourier-witness definition) and where $\epsilon$ is very small, the $1/\delta$ term dominates the complexity, while the remaining terms are independent of $\epsilon$. Getting tighter upper and lower bounds is a very interesting question. Interestingly, this question remains unresolved even for the simple example of the standard gaussian distribution in one dimension, which was investigated in prior work.
>
> **Regarding Example Distributions Handled by Prior Work**: The special cases previously understood were the Gaussian and Laplace settings, as discussed in the paper (lines 71-82). The uniform distribution example is new and is investigated in this work. The techniques used there are also based on Fourier analysis and deconvolution, and our upper bound can be viewed as a generalization of this approach to arbitrary base distributions. In particular, rather than relying on distribution-specific structure, our algorithm uses the Fourier-witness criterion to identify informative frequency regions and then searches over candidate means accordingly, as reflected in the algorithmic procedure. On the lower-bound side, prior techniques were only available in the Gaussian setting (by [Kotelak and Gao 2025]); as we note in the Remark 4.7 in the paper, our lower-bound construction differs in an essential way in order to handle arbitrary distributions. We refer the reviewer to our response to Reviewer Evgq for more details on the differences from [Kotelak and Gao 2025].

---

> > ### Author Rebuttal · Reviewer_4Etp · 2026-04-02
> >
> > Thank you for your response! I will keep my positive score as is.

---

### Official Review · Reviewer_Sm1w · 2026-03-13

**Soundness:** 4
**Presentation:** 4
**Significance:** 3
**Originality:** 3
**Overall Recommendation:** 5
**Confidence:** 3

**Summary:**

This paper studies the problem of mean estimation under mean-shift contamination.
In the mean-shift contamination model, samples are independently generated from $(1-\alpha)(D_\mu) + \alpha D_z$ where $\mu$ is the fixed true mean, $D_\mu$ is the "clean" distribution with mean $\mu$, $z$ is drawn from an arbitrary distribution, $D_z$ is the "adversarial" distribution with mean $z$.

Previous works have studied the specific cases when $D$ is Gaussian or Laplace.
This paper characterizes the sample complexity of this problem for general $D$.
Specifically, for distribution $D$ that satisfies several technical conditions, given corruption rate $\alpha$ and target error $\epsilon$, they define a quantity $\delta = \delta(\epsilon, \alpha, D) := \inf_{\|v\| \ge \epsilon} \sup_{\omega : dist(\omega \cdot v, \Z) \ge \alpha} |\phi_D(\omega)|$, and shows that a sample complexity upper bound of roughly $d/\delta^2$ and a lower bound of $(1/\delta)^{\Omega(1)}$ for $d=1$.
Their techniques make heavy use of Fourier analysis.

**Compliance With Llm Reviewing Policy:**

Affirmed.

**Final Justification:**

The rebuttal addressed my main concerns.
This paper presents nicely interesting and solid results and has no obvious weakness.
The only suggestion for the authors is to put more efforts on motivating the mean-shift contamination model, as this model is not so well-known and might look a bit unnatural for people who see it for the first time.
Overall, I recommend acceptance.

**Key Questions For Authors:**

- I'm curious about the applicability/generality of your results. Could you say something more about the application of your results to common distributions? E.g., sub-gaussian, sub-exponential, log-concave, etc.
- The assumption that the algorithm needs to access phi_D or sampling oracle for D could be a bit restrictive in some cases. Could you say something about what can we do if we only have sample access to the corrupted distribution?

**Limitations:**

yes

**Strengths And Weaknesses:**

// strengths
- The contamination model itself is somewhat interesting, but its underlying motivation, i.e. when is consistent estimation possible, makes lots of sense to me.
- The use of characteristic functions to characterize sample complexity in robust estimation seems novel.
- This is a well-written paper.

// weakness
- Technically, the mean-shift contamination model could be interesting to study, but conceptually the model looks a bit unnatural to me. Although the authors tried to make the point that this model is well-studied by citing other works, but it is still not very clear to me why this is an interesting model unless I dive into those works myself. I would suggest adding more discussions on how the mean-shift contamination model is connected to those works.

---

> ### Author Rebuttal · Authors · 2026-03-31
>
> We thank the reviewer for their careful reading and for the positive assessment of our paper. We are especially glad that the reviewer found the problem well-motivated, the paper well-written, and the Fourier-based characterization novel.
>
> **Applicability for Broad Classes**: Regarding the reviewer’s question about applicability to broad classes such as sub-gaussian, sub-exponential, or log-concave distributions, an important point is that our characterization is not governed by the tail behavior of the distribution, but rather by the tail behavior of its characteristic function. In other words, the relevant quantity is the Fourier witness, which depends much more directly on the Fourier decay structure—and therefore on the smoothness properties of the underlying density—than on tail classes alone. For this reason, one should not expect a tight classification stated only in terms of being sub-gaussian or sub-exponential. This is already reflected in Table 1: for example, both the Gaussian and the uniform distribution are sub-gaussian, yet they exhibit qualitatively very different sample complexities in our setting. More specifically, the Gaussian case has complexity exponential in $(\alpha/\epsilon)^2$, whereas for the uniform distribution the complexity is only polynomial.
>
> **Model Motivation**:
> We refer the reviewer to the response to reviewer Evgq for model motivation as discussed in the paper. A detailed discussion regarding model motivation and related work done in the work [Diakonikolas et al., 2025].
>
> **Estimation with only Corrupted Samples**:
> The problem is information-theoretically impossible without some knowledge about the base distribution. The reason is that when receiving samples from $D * Q$ we wouldn’t know if the base distribution was $D$ or $D * Q$ and they can have arbitrarily different means.
>
> [Diakonikolas et al., 2025] Efficient Multivariate Robust Mean Estimation Under Mean-Shift Contamination, Ilias Diakonikolas, Giannis Iakovidis, Daniel M. Kane, Thanasis Pittas, ICML 2025

---

> > ### Author Rebuttal · Reviewer_Sm1w · 2026-04-01
> >
> > Thanks for the response.
> > I'd like to clarify that, I think the motivation question---when is consistent estimation possible---makes sense, but I'm still not sure about the mean-shift contamination model studied by this paper.

---

### Official Review · Reviewer_DeA6 · 2026-03-15

**Soundness:** 3
**Presentation:** 3
**Significance:** 4
**Originality:** 4
**Overall Recommendation:** 6
**Confidence:** 3

**Summary:**

This work studies the work of mean estimation but when a constant fraction of the samples can be corrupted. They specifically consider the mean contamination model where the corrupted samples are obtained by adding a random variable drawn from a distribution which does have the same mean, hence “corrupting” the mean of the sample. They obtain (almost) matching upper and lower bounds for the sample complexity of the task. Their main contribution is to introduce the concept of Fourier witness which characterizes the sample complexity depending on the distribution.

**Compliance With Llm Reviewing Policy:**

Affirmed.

**Key Questions For Authors:**

NA

**Limitations:**

Yes

**Strengths And Weaknesses:**

Strengths -

1. The paper is well-written and sound.
2. They manage to essentially settle the general problem of characterizing the sample complexity for arbitrary distributions (error) while prior work focused on solving the problem for specific distributions.
3. That they have a matching lower using the exact same “Fourier-witness” is  impressive, as it essentially settles the problem.

Weakness -
1. The paper is complete but a bit dense regarding the technical content.
2. The only issue I have is that the authors did not give an explanation or any idea as to what the Fourier transform is and why it is relevant in this case. It is also not clear, for example, if this work is the first one to make use the "Fourier-witness” criteria. Additionally, as a reader, it will help me understand and appreciate it more without understanding all the proofs.
3. There are no experiments done. Personally I feel that experiments, in this case would be really easy to do and at the same time add a lot to the content of the paper.

---

> ### Author Rebuttal · Authors · 2026-03-31
>
> First, we thank the reviewer for the careful reading and for the positive assessment of our work. We are especially glad that the reviewer found the paper well-written and sound, and that the reviewer appreciated the fact that our upper and lower bounds are both governed by the same Fourier-witness quantity. We also appreciate the reviewer’s suggestion that the paper could give more intuition for the role of Fourier analysis and for the meaning of the Fourier-witness criterion.
>
> **Role of the Fourier transform**:  We point to the parts of our submission that provide intuition of why the Fourier transform—and the notion of the Fourier witness—are essential in characterizing the problem. Specifically, as explained in the technical overview section (lines 145–155), the observed distribution has the convolution form $D_{\mu}^{(\alpha)}=D∗((1-\alpha)\delta_{\mu}+\alpha Q)$ in the mean-shift contamination model, and therefore its characteristic function factors as $\phi_{D_{\mu}^{(\alpha)}}=\phi_D((1-\alpha)\phi_{\delta_{\mu}}+\alpha\phi_Q)$.
> This makes the Fourier domain natural, since it turns the complicated convolution operation into a simple multiplication operator. In particular, the base distribution D shared by the inlier and outlier parts of the model becomes a common multiplicative factor $\phi_D$ after the deconvolution step. This also explains why the magnitude of $\phi_D​$ is crucial: if $\phi_D(\omega)$ is very small, then this deconvolution becomes unstable and statistically costly. After deconvolution, one is essentially comparing the clean mean signal $(1-\alpha)e^{2\pi i \mu \cdot \omega}$ against the outlier/noise signal $\alpha \phi_Q(\omega)$. Thus, to distinguish an incorrect candidate $\hat{\mu}$ from the true $\mu$, one needs frequencies where this difference is detectable, i.e., where $(\hat{\mu}-\mu)\cdot \omega$ is $\alpha$-far from an integer, and where the Fourier mass is still non-negligible. This is exactly the role of the Fourier witness, as described in (lines 115-134, right column) and formalized in Definition 3.1. On the lower-bound side, the same phenomenon appears in its complementary form: the hard instances are precisely those where the Fourier mass is concentrated near the “bad” band regions, as discussed in (Remark 4.3).
>
> **Experimental Results**:  We would like to thank the reviewer for this suggestion and we agree that numerical illustrations could be interesting. This is something that we plan to pursue as a future direction. That said, several of the prior works we cite on this model and related deconvolution-based settings already include numerical investigations, whereas our focus here is different: to give a tight theoretical characterization of the sample complexity for general base distributions, including both upper and lower bounds. In this sense, the main goal of the paper is to settle the information-theoretic question rather than to optimize a practical estimator.

---

> > ### Author Rebuttal · Reviewer_DeA6 · 2026-04-03
> >
> > Thank you for the rebuttal. I will keep my score as is but urge the authors to add the intuition to the writeup. And more importantly add an explicit section for related works discussing prior works especially of Kotekal and Gao.

---

### Official Review · Reviewer_Evgq · 2026-03-23

**Soundness:** 3
**Presentation:** 3
**Significance:** 3
**Originality:** 3
**Overall Recommendation:** 4
**Confidence:** 3

**Summary:**

This paper considers the problem of robust mean estimation. The robustness model is different from most of the previous work and is as follows: there exists some true mean $\mu$, some true mean zero distribution $D$, and some arbitrary corruption distribution $Q$, and then each sample with probability $1-\alpha$ is drawn from a shift of $D$ by $\mu$ and with probability $\alpha$ it's drawn from $D \ast Q$. This adversary is weaker than the Huber contamination model it is only allowed to sample the corrupted samples from convolutions of $D$, whereas in the Huber contamination model it is allowed to sample from arbitrary distributions. In particular, in the Huber contamination model it is impossible to get arbitrarily close to the true mean by taking more samples (perform consistent estimation), whereas here it is possible.

Previous work of (Kotekal and Gao, 2025) gave the minimax rates for the case where $D$ is a Gaussian, and leave the problem of general $D$ open. This work characterizes the rates in terms of the Fourier transform (characteristic function) of the distribution $D$. In particular they show that the "right" object that characterizes learnability up to accuracy $\epsilon$ under $\alpha$ fraction corruption, is the minimum (worst case) over $v$ such that $\lVert{v}\rVert \ge \epsilon$, largest Fourier magnitude $\phi_D(w)$ such that $\langle w, v \rangle$ stays away by $\alpha$ from all integers. They name this quantity $\delta(\epsilon, \alpha, D)$ and show that $\text{poly}(d, 1/\delta)$ many samples are necessary and sufficient for estimation under $\alpha$ corruption and $\epsilon accuracy.

Technically, the argument for the upper bound is a cover-based argument: Pick a cover over the space of $\mu$ and pick a cover over the space of $w$, then the problem becomes finding a $\hat{\mu}$ that does not have a $w$ that witnesses it's inaccuracy. In order to do this for a fixed $\hat{\mu}$, it suffices to compare the empirical $\phi_Q(w)$ against $\exp(2\pi i \langle \hat{w}, w \rangle)$ to see if it leads to a large difference.

**Compliance With Llm Reviewing Policy:**

Affirmed.

**Final Justification:**

I maintain my score. My main remaining reservation is the model's specialized nature, which limits the paper’s broader impact. Otherwise, I find the paper technically solid, clearly presented, and novel, and the rebuttal resolved my main concerns.

**Key Questions For Authors:**

None for now.

**Limitations:**

yes.

**Strengths And Weaknesses:**

Soundness: The proofs have not been checked in detail, but at a high level the strategy and techniques all seem reasonable. No major issues found.

Presentation: The presentation has been done well.

Significance: I understand the model is an interesting theoretical model to study from a theoretical perspective but It is not very clear to me where the motivation to study this model comes from. The authors also have not mentioned any cases in the introduction where this model might relate to some real world phenomenon. That being said, the problem was mentioned in prior work as an interesting problem to consider, so there would be some interest in the robust statistics community in this work.

Originality: The characterization is novel. They characterize the minimax rates in this model under arbitrary distributions $D$ which was not fully understood prior to this work. In terms of the techniques (Kotekal and Gao, 2025) already use Fourier analysis to obtain similar results for Gaussian mean estimation. The technical novelty of this work is extending similar techniques to arbitrary distributions.

---

> ### Author Rebuttal · Authors · 2026-03-31
>
> We thank the reviewer for their time and effort spent reviewing our work. We respond to the points brought up by the reviewer in turn.
>
>
> **Motivation and Significance of the Mean-Shift Model**:
> We provide discussion on the motivation and significance of the model in the introduction (lines 58-70). In particular, as noted there, this contamination model originates from the multiple-hypothesis-testing literature, where the parameters of the null distribution are not known a priori and must be estimated from the data. We would also like to mention that this model has been used as a modeling assumption for large-scale microarray testing in Efron’s work [Efron 2008], which provides a concrete setting in which this model naturally arises.
>
>
> **Comparison to [Kotekal and Gao, 2025]**:
> While both works use Fourier analysis for both upper and lower bounds, we would like to emphasize that our approach is substantially different. First, our upper bound uses fundamentally weaker assumptions of the existence of Fourier witnesses that *characterizes* the sample complexity of the problem for a general distribution. Additionally, our work addresses the high-dimensional case, whereas Kotelak and Gao consider the 1-dimensional case.  At a technical level, Kotekal and Gao establish their lower bound by enforcing matching of the characteristic functions in an interval near the origin, which suffices for the gaussian case. However, for general base distributions, matching only near the origin does not suffice. Our lower bound is based on a more general idea, where we instead match periodically across the entire collection of usable frequency bands—namely the frequencies relevant for the estimation algorithm. This step requires a new smooth periodized window argument. This is exactly the reason why our construction cancels the discrepancy on all relevant segments, while still preserving realizability by distributions and the tail control needed for the TV bound. We make this distinction explicit in Remark 4.7.
>
>
> [Efron 2008], Efron B. Microarrays, empirical bayes and the two-groups model. 2008.

---

> > ### Author Rebuttal · Reviewer_Evgq · 2026-04-04
> >
> > Thank you to the authors for the clear rebuttal. It addresses my main concerns. I maintain my positive evaluation.

---

### Decision · Program_Chairs · 2026-04-30

**Decision:**

Accept (regular)

**Comment:**

The paper proposes a new algorithm for mean estimation under mean-shift contamination. The key difference established by the paper is that in this setting, with a large enough sample size, any target accuracy can be achieved, a separation from the Huber contamination setting typically studied. All of the reviewers agreed that the theoretical novelty is strong and the presentation quality is high. The main concerns are simply regarding the applicability of the model: while the authors mention some motivating applications, the contamination model is much more restrictive than prior analogous works, and the reviewers note that this motivation could be more self-contained in the paper. Overall I think that both the results and techniques will be of interest to a broad audience and recommend acceptance.